# NeuroLifting: Neural Inference on Markov Random Fields at Scale

## Abstract

Inference in large-scale Markov Random Fields (MRFs) is a critical yet challenging task, traditionally approached through approximate methods like belief propagation and mean field, or exact methods such as the Toulbar2 solver. These strategies often fail to strike an optimal balance between efficiency and solution quality, particularly as the problem scale increases. This paper introduces NeuroLifting, a novel technique that leverages Graph Neural Networks (GNNs) to reparameterize decision variables in MRFs, facilitating the use of standard gradient descent optimization. By extending traditional lifting techniques into a non-parametric neural network framework, NeuroLifting benefits from the smooth loss landscape of neural networks, enabling efficient and parallelizable optimization. Empirical results demonstrate that, on moderate scales, NeuroLifting performs very close to the exact solver Toulbar2 in terms of solution quality, significantly surpassing existing approximate methods. Notably, on large-scale MRFs, NeuroLifting delivers superior solution quality against all baselines, as well as exhibiting linear computational complexity growth. This work presents a significant advancement in MRF inference, offering a scalable and effective solution for large-scale problems.

## 1 Introduction

Markov Random Fields (MRFs) stand as a fundamental computational paradigm for modeling complex dependencies among a large collection of variables, permeating a variety of domains such as computer vision (Wang et al., 2013; Su et al., 2021), natural language processing (Almutiri & Nadeem, 2022; Ammar et al., 2014; Lin et al., 2020), and network analysis (Wu et al., 2020; Yunfei Ma & Razavi, 2022). MRF's capacity to encode intricate probabilistic interactions underscores its widespread utility. However, unraveling the optimal configurations in high-dimensional settings remains a formidable task owing to the inherent computational complexity involved.

Traditional inference methodologies for MRFs bifurcate into approximate and exact strategies, each with its own set of advantages and limitations. Approximate inference techniques, such as belief propagation (Pearl, 2022; Wainwright et al., 2005) and mean field (Saito et al., 2012; Zhang, 1993) approximations, strive for computational efficiency but often at the expense of solution quality, particularly as the scale of the problem escalates. Conversely, exact inference methods, epitomized by the Toulbar2 solver (De Givry, 2023; Hurley et al., 2016), aspire to optimality but are frequently hampered by exponential time complexities that render them infeasible for large-scale MRFs.

Despite significant advances, achieving a harmonious balance between efficiency and solution quality in large-scale MRF inference remains a largely unmet challenge. This paper addresses this pivotal issue through the introduction of "NeuroLifting" – a neural-network-driven paradigm that extends traditional lifting technique in the context of optimization (Albersmeyer & Diehl, 2010; Balas & Perregaard, 2002; Bauermeister et al., 2022). NeuroLifting is a novel approach that reimagines MRF inference by leveraging the potency of Graph Neural Networks (GNNs) alongside gradient-based optimization techniques.

The core innovation of NeuroLifting lies in the reparameterization of the decision variables within MRFs utilizing a randomly initialized GNN. While some recent heuristics succeeded in utilizing GNNs for solving combinatorial problems (Cappart et al., 2023; Schuetz et al., 2022), an effective adaptation to MRF inference remains opaque. Besides, they generally lack an in-depth

understanding of how GNNs facilitate downstream computation. In this paper, we for the first time bridge such practice to traditional lifting techniques, and further demonstrate that by harnessing the continuous and smooth loss landscape intrinsic to neural networks, NEUROLIFTING simplifies the optimization process for large-scale MRFs, enabling enhanced parallelization and performance on GPU devices.

Empirical evaluations substantiate the efficacy of NEUROLIFTING, showcasing its ability to deliver high-quality solutions across diverse MRF datasets. Notably, it outperforms all existing approximate inference strategies in terms of solution quality without sacrificing computational efficiency. When juxtaposed with exact strategies, NEUROLIFTING demonstrates comparable solution fidelity while markedly enhancing efficiency. For particularly large-scale MRF problems, encapsulating instances with over 50,000 nodes, NEUROLIFTING exhibits a linear computational complexity increase, paired with superior solution quality relative to exact methods.

In summary, the contributions of this paper are threefold. 1) **Methodical design**: we present NEUROLIFTING as an innovative and practical solution to the enduring challenge of efficient and high-quality inference in large-scale MRFs; 2) **Non-parametric lifting**: we extend the concept of lifting from traditional optimization practices into a modern neural network framework, thereby offering a fresh lens through which to tackle large-scale inference problems; 3) **Significant performance**: NEUROLIFTING achieved significant performance improvement over existing methods, showing remarkable scalability and efficiency in real-world scenarios.

## 2 PRELIMINARY

**Markov Random Field**. An MRF is defined over a undirected graph $\mathcal{G} = (\mathcal{V}, \mathcal{C})$, where $\mathcal{V}$ represents the index set of random variables and $\mathcal{C} \subseteq 2^{\mathcal{V}}$ is the clique set representing the (high-order) dependencies among random variables. Throughout this paper, we associate a node index $i$ with a random variable $x_i \in \mathcal{X}$, where $\mathcal{X}$ is a finite alphabet. Thus, given graph $\mathcal{G}$, the joint probability of a configuration of $X = \{x_i\}_{i \in \mathcal{V}}$ can be expressed as

$$\mathbb{P}(X) = \frac{1}{Z} \exp(-E(X)) = \frac{1}{Z} \exp\left( -\sum_{i \in \mathcal{V}} \theta_i(x_i) - \sum_{C_k \in \mathcal{C}} \theta_{C_k}(\{x_l | \forall x_l \in C_k\}) \right) \quad (1)$$

where $Z$ is the partition function, $\theta_i(\cdot)$ denotes the unary energy functions, $\theta_C(\cdot)$ represent the clique energy functions. In this sense, MRF provides a compact representation of probability by introducing conditional dependencies:

$$\mathbb{P}(x_i | X \backslash \{x_i\}) = \mathbb{P}(x_i | \{x_j\} \text{ for } i, j \in C_k \text{ for } C_k \in \mathcal{C}). \quad (2)$$

In this paper, we consider the Maximize a Posterior (MAP) estimate of Equation 1, which requests optimizing Equation 1 via $X^* = \min_X E(X)$. One can consult Koller & Friedman (2009) for more details.

**Graph Neural Networks**. GNNs represent a distinct class of neural network architectures specifically engineered to process graph-structured data (Kipf & Welling, 2017; Hamilton et al., 2017; Xu et al., 2019; Veličković et al., 2018). In general, when addressing a problem involving a graph $\mathcal{G} = (\mathcal{V}, \mathcal{E})$, where $\mathcal{E}$ is the edge set, GNNs utilize both the graph $\mathcal{G}$ and the initial node representations $\{h_i^{(0)} \in \mathbb{R}^d | \forall i \in \mathcal{V}\}$ as inputs, where $d$ is the dimension of initial features. Assuming the total number of GNN layers to be $K$, at the $k$-th layer the graph convolutions typically read:

$$h_i^{(k)} = \sigma\left( W_k \cdot \text{AGGREGATE}^{(k)} \left( \left\{ h_j^{(k-1)} : j \in \mathcal{N}(i) \cup \{i\} \right\} \right) \right) \quad (3)$$

where $\text{AGGREGATE}^{(k)}$ is defined by the specific model, $W_k$ is a trainable weight matrix, $\mathcal{N}(i)$ is the neighborhood of node $i$, and $\sigma$ is a non-linear activation function, e.g., ReLU.

**Optimization with Lifting**. Lifting is a sophisticated technique employed in the field of optimization to address and solve complex problems by transforming them into higher-dimensional spaces (Balas, 2005; Papadimitriou & Steiglitz, 1982). By introducing auxiliary variables or constraints, lifting serves to reformulate an original optimization problem into a more tractable or elucidated form, often making the exploration of optimal solutions more accessible. In the context of MRFs, lifting can be utilized to transform inference problems into higher dimensions where certain

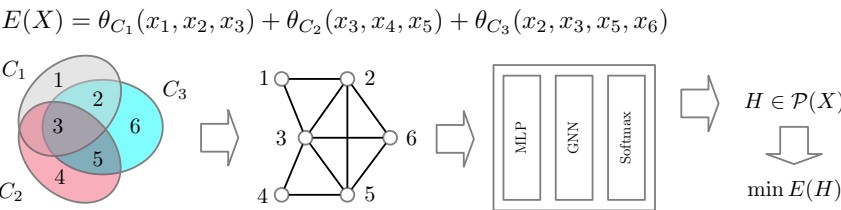

$$E(X) = \theta_{C_1}(x_1, x_2, x_3) + \theta_{C_2}(x_3, x_4, x_5) + \theta_{C_3}(x_2, x_3, x_5, x_6)$$

Figure 1: An overview of NEUROLIFTING.

properties or symmetries associated with specific MRF problems are more easily exploitable (Wainwright et al., 2005; Globerson & Jaakkola, 2007; Bauermeister et al., 2022). However, a principled lifting technique is still lacking for generalized MRFs.

## 3 METHODOLOGY

### 3.1 OVERVIEW

An overview of NEUROLIFTING is in Figure 1, with an exemplary scenario involving an energy function devoid of unary terms, yet comprising three clique terms. Initially, the clique-based representation of this function (depicted in the leftmost shaded diagram) undergoes a transformation to a graph-based perspective, which subsequently integrates into the network architecture. To address the absence of inherent node feature information in the original problem, we elevate the dimensionality of decision variables within this framework. This transformation facilitates a paradigm shift from the identification of optimal state values to the learning of optimal parameters for encoding and classification of these variables. Furthermore, we devised a novel approach to circumvent the absence of a traditional loss function, thereby extending the applicability of our framework to Markov Random Fields (MRFs) of arbitrary order.

### 3.2 PREPOSSESSING

We discuss several necessary preprocessing steps to adapt standard MRF to a GNN style.

**Topology construction for GNNs.** In an MRF instance, the high-order graph structure consists of nodes and cliques, diverging from typical GNNs allowing only pairwise edges (2nd-order). To facilitate the power of GNNs, we need to convert high-order graph into a pairwise one. By the very definition of a clique, any two nodes that appear within the same clique are directly related. Thus, for any two nodes $i, j \in C_k$ in a clique $C_k$, we add a pairwise edge $(i, j)$ to its GNN-oriented graph. An example can be observed in Figure 1. It is worth noting that an edge may appear in multiple cliques; however, we add each edge only once to the graph.

**Initial feature for GNNs.** As there is no initial features associated to MRF instances, we initialize feature vectors to GNNs *randomly* with a predefined dimension $d$. Detailed information on how we will handle these artificial features to ensure they effectively capture the underlying information of the problem will be provided in Section 3.3.

**Vectorizing the energy function.** The transformed energy function $E(X)$ will serve as the loss function guiding the training of the neural networks. In Section 3.4, we will detail the transformation process and discuss how to effectively utilize it. Note the values of these functions can be pre-evaluated and repeatedly used during the training process. Therefore, we employ a look-up table to memorize all function values with discrete inputs. For unary energies, we denote the vectorized unary energy of variable $x_i$ as $\phi(x_i)$, where the $n$-th element corresponds to $\theta_i(x_i = n)$. Similarly, we represent the clique energy for clique $C_k$ using the tensor $\psi(\{x_l | \forall x_l \in C_k\})$. This tensor can be derived using the same conceptual framework; for instance, the element $\psi(x_i, x_j, x_k)$ at position $(0, 2, 4)$ corresponds to the value of $\theta_{\{i,j,k\}}(x_i = 0, x_j = 2, x_k = 4)$ .

**Padding node embeddings & energy terms.** GNNs typically require all node embeddings to be of the same dimension, meaning that the embeddings $h^{(K)}$ at $K$-th layer must share the same size. However, in general MRFs, the variables often exhibit different numbers of states. While traditional

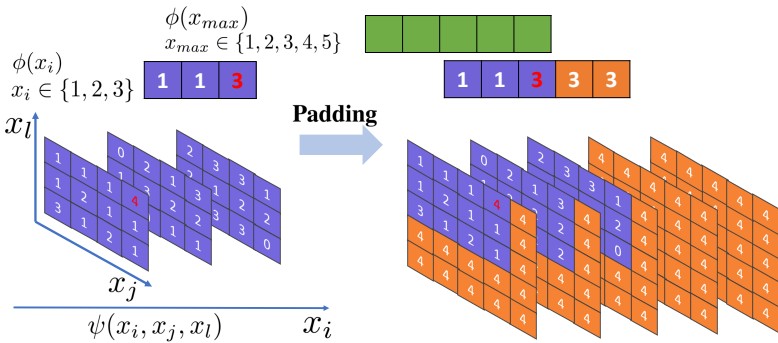

Figure 2: This illustrates the padding procedure for unary loss terms $\phi(x)$ and clique loss terms $\psi(x_i, x_j, x_k)$, with $|\mathcal{X}| = 5$. $x_{max}$ denotes the variable that has the maximum value range. The elements shown in purple represent the energy values in the original $\phi$ and $\psi$. After padding, the dimension of vector $\phi$, as well as each dimension of the energy tensor $\psi(x_i, x_j, x_k)$, will be 5. The padded portion is indicated in orange, with values either $\max(\phi)$ or $\max(\psi)$.

belief-propagation-based methods can easily manage such variability, adapting GNNs to handle these discrepancies is less straightforward.

To address this mismatch, we employ padding strategy – a common technique used to handle varying data lengths. This strategy is applied to both the node embeddings and the unary and pairwise (or clique) energies, to ensure consistent embedding dimensions. Concretely, we assign virtual states to the nodes whose state number is less than $|\mathcal{X}|$. Then, we assign energies to those padded labels with the *largest value of the original energy term*. The schematic diagram of the padding procedure is in Fig. 2. In this example, we consider the case where $|\mathcal{X}| = 5$. We start with the unary energy vector for $x_i$ denoted as $\phi(x_i) = \{1, 1, 3\}$, which has three states. Before padding, the highest value in this vector is 3, highlighted in red, and this value will be used for padding. The padded vector is shown on the right-hand side of the figure, with the padded portion indicated in orange. For the clique terms, we will apply padding similarly to the unary terms. The original energy matrix for the clique involving nodes $i, j, l$ has a dimension of $3 \times 3 \times 4$. Given that $|\mathcal{X}| = 5$, we need to pad the matrix so that $\psi(x_i, x_j, x_l) \in R^{5 \times 5 \times 5}$. In this case, the largest value in the original energy matrix is 4. As depicted in the figure, all padded values in the orange area are filled with 4. This approach of assigning high energies to the padded labels serves to discourage the model from selecting these padded states, thereby incentivizing it to choose the original, non-padded states with lower energies.

*Remark.* Other strategies are also being considered. If the padded energies are set to the largest element among all original energies or to a significantly larger value compared to the original values, this approach can dramatically alter the loss landscape. As a result, the model may converge to an infeasible point in the original problem, leading it to select padded states instead. A similar issue arises when we mask the padded regions during loss calculation. This masking operation can introduce significant interference in the optimization process, preventing the model from achieving a high-quality feasible solution.

### 3.3 GNNs as Non-parametric Lifting

In this section, we detail how NEUROLIFTING generates features that capture the hidden information of the given MRF and solves the original MAP problem by optimizing in a high-dimensional parameter space. As mentioned in Section 3.2, we initially generate *learnable* feature vectors randomly using an encoder that embeds all nodes, transforming the integer decision variables into $d_l$-dimension vectors $h_i^{(0)} \in \mathbb{R}^{d_l}$ for node $i$, where $d_l$ is a hyperparameter representing the dimension after lifting.

The intuition for utilizing GNNs in the implementation of lifting techniques is inspired by Loopy Belief Propagation (LBP) (Weiss & Freeman, 2001). When applying LBP for inference on MRFs, the incoming message $M_{ji}$ to node $i$ from node $j$ is propagated along the edges connecting them.

Node $i$ can then update its marginal distributions according to the formula in Eq. 4.

$$p^{\text{posterior}}(x_i|X\backslash\{x_i\}) = p^{\text{prior}}(x_i|X\backslash\{x_i\}) \prod_{(i,j)\in\mathcal{E}} \sum_{x_j} M_{ji} \tag{4}$$

Importantly, the incoming messages are not limited to information solely about the directly connected nodes; they also encompass information from sub-graphs that node $i$ cannot access directly without assistance from its neighbors. This allows a more comprehensive aggregation of information, enabling node $i$ to merge these incoming messages with its existing information. This process of message aggregation bears resemblance to the message-passing procedure used in GNNs, where nodes iteratively update their states based on the information received from their neighbors.

Graph convolutions should intuitively treat adjacent nodes equally, consistent with the principle in MRFs, where the information collected from neighbors is processed equally. Typical GNNs are summarized in the followings:

| | Graph Convolutions | Neighbor Influence |
|---|---|---|
| GCN | $h_i^{(k)} = \sigma\left(W_k \cdot \sum_{j\in\mathcal{N}(i)\cup\{i\}}(\deg(i)\deg(j))^{-1/2}h_j^{(k-1)}\right)$ | **Unequal** |
| GAT | $h_i^{(k)} = \sigma\left(\sum_{j\in\mathcal{N}(i)\cup\{i\}}\alpha_{i,j}W_k h_j^{(k-1)}\right)$ | **Unequal** |
| GraphSAGE | $h_i^{(k)} = \sigma\left(W_k \cdot h_i + W_k \cdot (|\mathcal{N}(i)|)^{-1}\sum_{j\in\mathcal{N}(i)}h_j^{(k-1)}\right)$ | **Equal** |

where $\deg(i)$ is the degree of node $i$, $\alpha_{i,j}$ is the attention coefficients, and $|\mathcal{N}(i)|$ is the neighborhood size of node $i$. According to the influence of neighbors, they can be classified into three categories: 1) neighborhood aggregation with normalizations (e.g., GCN (Kipf & Welling, 2017) normalize the influence by node degrees), 2) neighborhood aggregation with directional biases (e.g., GAT (Veličković et al., 2018) learn to select the important neighbors via an attention mechanism), and 3) neighborhood aggregation without bias (e.g., GraphSAGE (Hamilton et al., 2017) directly aggregate neighborhood messages with the same weight). Therefore, we select the aggregator in GraphSAGE as our backbone for graph convolutions. The performance of these GNN backbones on our MRF datasets is shown in Fig. 3.

Another primary characteristic of MRFs is its ability to facilitate information propagation across the graph through local connections. This means that even though the interactions are defined locally between neighboring nodes, the influence of a node can extend far beyond its immediate vicinity. As a result, MRFs can effectively capture global structure and dependencies within the data. We thus use Jumping Knowledge (Xu et al., 2018) to leverage different neighborhood ranges. By doing so, features representing local properties can utilize information from nearby neighbors, while those indicating global states may benefit from features derived from higher layers.

At each round of iterations, we optimize both the GNN parameters and those of the encoder. At the start of the next iteration, we obtain a new set of feature vectors, $\mathcal{H}_t^{(0)} = \{h_{i,t}^{(0)} \in \mathbb{R}^{d_l}|\forall i \in \mathcal{V}\}$, where $t$ indicates the $t$-th iteration. This process enables us to accurately approximate the latent features of the nodes in a higher-dimensional space.

### 3.4 ENERGY MINIMIZATION WITH GNN

As indicated by Equation 1, the energy function can serve as the loss function to guide network training since minimizing this energy function aligns with our primary objective. Typically, the energy function for a new problem instance takes the form of a look-up table, rendering the computation process non-differentiable. To facilitate effective training in a fully unsupervised setting, it is crucial to transform this computation into a differentiable loss aligning with the original energy function.

The initial step involves transforming the decision variable from $x_i \in \{1,...,s_i\}$, where $s_i$ is the number of states of variable $x_i$, to $v_i \in \{0,1\}^{s_i}$. At any given time, exactly one element of the vector $v_i$ can be one, while all other elements must be zero; the position of the 1 indicates the current state of the variable $x_i$. Define $V_k = \otimes_{i\in C_k}v_i$, where $\otimes$ is the tensor product. The corresponding energy function would be Equation 5. Subsequently, we relax the vector $v_i$ to $p_i(\theta) \in [0,1]^{s_i}$, where $p_i(\theta)$ represents the output of our network and $\theta$ denotes the network parameters. This output can be interpreted as the probabilities of each state that the variable $x_i$ might assume.

$$E(\{v_i|i \in \mathcal{V}\}) = \underbrace{\sum_{i\in\mathcal{V}}\langle v_i(\theta), \phi(x_i)\rangle}_{\text{Unary Term}} + \underbrace{\sum_{C_k\in\mathcal{C}}\langle\psi(C_K), V_k\rangle}_{\text{Clique Term}} \tag{5}$$

$$L(\theta) = \underbrace{\sum_{i \in \mathcal{V}} \langle p_i(\theta), \phi(x_i) \rangle}_{\text{Unary Term}} + \underbrace{\sum_{C_k \in \mathcal{C}} \langle \psi(C_K), P_k \rangle}_{\text{Clique Term}} \tag{6}$$

where $\langle \cdot, \cdot \rangle$ refers to the tensor inner product. The applied loss function is defined in Equation 6, here $P_k = \otimes_{i \in C_k} p_i$. The rationale behind our loss function closely resembles that of the cross-entropy loss function commonly used in supervised learning. Let $P$ represent the true distribution and $Q$ denote the predicted distribution. A lower value of cross-entropy $H(P, Q)$ indicates greater similarity between these two distributions. However, our approach differs in that we are not seeking a predicted distribution that closely approximates the true distribution. Instead, for each variable, we aim to obtain a probability distribution that is highly concentrated, with the concentrated points corresponding to the states that minimize the overall energy.

Once the network outputs are available, we can easily determine the assignments by *rounding* the probabilities $p(\theta)$ to obtain binary vectors $v$. Using these rounded results, the actual energy can be calculated using Equation 5. It is observed that after the network converges, the discrepancy between $L(\theta)$ and $E(\{v_i | i \in \mathcal{V}\})$ is minor and we won't see any multi-assignment issue in decision variables. We choose Adam (Kingma & Ba, 2015) as the optimizer, and employ simulated annealing during the training process, allowing for better exploring the loss landscape to prevent sub-optima.

### 3.5 ANALYSIS AND DISCUSSION

**Relation to lifting.** In this innovative framework of using GNNs for inference on MRFs, a natural and sophisticated parallel emerges with the classical concept of lifting in optimization (Balas et al., 1993). By mapping each unary term of an MRF to a node within a GNN and translating clique terms into densely connected subgraphs, the traditional MRF energy minimization transforms into optimizing a multi-layer GNN with extra dimensionality. This procedure aligns with the lifting technique where the problem space is expanded to facilitate more efficient computation. Akin to the principle of standard lifting to ease optimization, the GNN-based reparameterization can leverage the gradient descent optimization paradigm inherent in the smooth neural network landscape (Dauphin et al., 2014; Choromanska et al., 2015), ensuring efficient computation and convergence. Therefore, while offering an enhanced approach to inference, the GNN reparameterization mirrors the core principles of lifting by transforming and extending the solution space into a computation-friendly one to achieve computational efficacy and scalability. More empirical evidence is in Sec. 4.4.

**Complexity analysis.** The primary computations in this model arise from both the loss calculation and the operations within the GNN. For the loss function, let $c_{max}$ denote the maximum clique size. The time complexity for the loss calculation is given by $O(|\mathcal{V}||\mathcal{X}| + c_{max}|\mathcal{C}||\mathcal{X}|)$. For the GNN component, let $\mathcal{N}_v$ denote the average number of neighbors per node in the graph. The time complexity for neighbor aggregation in each layer is $O(\mathcal{N}_v|\mathcal{V}|)$, and merging the results for all nodes requires $O(|\mathcal{V}|d)$ where $d$ is the feature dimension. Thus, for a $K$-layer GraphSAGE model with the custom loss function, the overall time complexity can be expressed as $O(|\mathcal{X}|(|\mathcal{V}| + c_{max}|\mathcal{C}|) + K|\mathcal{V}|(\mathcal{N}_v + d))$. This analysis highlights the efficiency of the framework in managing large-scale graphs by leveraging neighborhood sampling and aggregation techniques. The derived complexity indicates that the model scales linearly with respect to the number of nodes, the number of layers, and the dimensionality of the feature vectors, making it well-suited for large-scale instances.

## 4 EXPERIMENT

**Evaluation metric.** For all instances used in the experiments, we utilize the final value of the overall energy function $E(X)$ as defined in Equation 1. Without loss of generality, all problems are formulated as minimization problems.

**Baselines.** We compare our approach against several well-established baselines: Loopy Belief Propagation (LBP), Tree-reweighted Belief Propagation (TRBP) (Wainwright et al., 2005), and Toulbar2 (Brouard et al., 2020). LBP is a widely used approximate inference algorithm that iteratively passes messages between nodes. TRBP improves upon LBP by introducing tree-based reweighting to achieve better approximations, particularly in complex graph structures. Toulbar2 is an exact optimization tool based on constraint programming and branch-and-bound methods Notably, Toulbar2

is the winner on **all** MPE and MMAP task categories of UAI 2022 Inference Competition [1]. These baselines allow us to evaluate the performance of our proposed solution under fair settings.

**MRF format and transformation.** The MRF data files are in UAI format and we interpret the data files in the same way as Toulbar2. Detailed information about unary and clique terms will be treated as unnormalized (joint) distributions, and the energies are calculated as $\theta_i(x_i = a) = -log(P(x_i = a))$, where $P(x_i = a)$ represents the probability provided by the data file. Note that we use the unnormalized values during the transformation process. The transformation for the clique energy terms will follow the same procedure. More details are in Appendix E.

### 4.1 Synthetic Problems

We first conduct experiments on synthetic problems generated randomly based on Erdős–Rényi graphs (Erdös & Rényi, 1959). The experiments are divided into pairwise cases and higher-order cases. We will compare the performance of NeuroLifting with LBP, TRBP, and Toulbar2 on pairwise MRFs. For the higher-order MRF cases, we will compare NeuroLifting exclusively with Toulbar2, as LBP and TRBP are not well-suited for handling the complexities inherent in high-order MRFs. The raw probabilities (energies) on the edges/cliques are randomly generated using the Potts function (Equation 7), representing two typical types found in the UAI 2022 dataset. The parameters $\alpha$ and $\beta$ serve as constant penalty terms and $\mathbb{I}$ is the indicator function.

$$\theta_{ij} = \alpha\mathbb{I}(x_i = x_j) + \beta \tag{7}$$

For all the random cases, all the probabilities values of the unary terms and pairwise (clique) terms are generated randomly range from 0.2 to 3.0. For the Potts models, $\alpha, \beta \in [0.00001, 1000]$. Each random node can select from 2 to 6 possible discrete labels, and the values of the unary terms are also generated randomly, ranging from 0.2 to 3.0. LBP and TRBP are allowed up to 60 iterations, with a damping factor 0.1 to mitigate potential oscillations. Toulbar2 operates in the default mode with time limit 18000s. We employ a 5-layer GNN to model all instances. The learning rate is set to $1e^{-4}$, and the model is trained for up to 150 iterations for each instance, utilizing a simple early stopping rule with an absolute tolerance of $1e^{-4}$ and a patience of 10. The data generation method and the parameter settings are the same for both pairwise cases and high order cases.

**Pairwise instances.** The inference results on pairwise cases are summarized in Table 1. Prefix "P_potts_" and "P_random_" indicate instances generated with Potts energy and random energy, respectively. It is evident that as the problem size scales up, NeuroLifting outperforms the baseline approaches; meanwhile, it also achieves comparable solution quality even when the problem sizes are small. This trend is consistent across both energy models.

**Higher-order instances.** The inference results on high oreder cases are summarized in Table 2. The "H" in the prefix stands for High-order and all the instances are generated using Potts model. The number of cliques in the table encompasses both the cliques themselves and the edges connecting them. The relationships between nodes are based on either pairwise interactions or clique relationships. We conduct tests on both a dense graph with a small size (H_Instances_1, H_Instances_2) and a sparse graph with a larger size. The results indicate that NeuroLifting outperforms Toulbar2 in both settings, demonstrating its ability to effectively handle not only large graphs but also dense graphs. This versatility highlights the robustness and effectiveness of NeuroLifting across different graph structures.

### 4.2 UAI 2022 Inference Competition datasets

We then evaluate our algorithm using instances from the UAI 2022 Inference Competition datasets, including both pairwise cases and high-order cases. The time settings will align with those established in the UAI 2022 Inference Competition, specifically 1200 seconds and 3600 seconds. LBP and TRBP algorithms are set to run for 30 iterations with a damping factor of 0.1, and the time limit for Toulbar2 is configured to 1200 seconds, which is generally sufficient for convergence. For NeuroLifting, we utilize an 8-layer GNN to model all instances, with the model trained for up to 100 iterations for each instance; other settings remain consistent with those used in the synthetic problems. We also experimented with lifting dimensions of 64, 512, 1024, 4096, and 8192.

---

[1]https://www.auai.org/uai2022/uai2022_competition

Table 1: Results on ER graphs with state numbers range from 2 to 6. Numbers out of the bracket correspond to the obtained energy values, the number in the brackets is the final loss given by the loss function. Best in bold.

| Graph | #nodes/#edges | LBP | TRBP | Toulbar2 | NEUROLLIFTING |
|---|---|---|---|---|---|
| P_potts_1 | 1k/7591 | -22215.700 | -21365.800 | **-22646.529** | -21451.025 |
| P_potts_2 | 5k/37439 | **-111319.000** | -105848.000 | -110022.248 | -105952.531 |
| P_potts_3 | 10k/75098 | **-221567.000** | -210570.000 | -218311.424 | -209925.269 |
| P_potts_4 | 50k/248695 | 12411.200 | 13454.600 | 12955.129 | **11679.429** |
| P_potts_5 | 50k/249624 | 25668.500 | 35389.000 | 12468.172 | **11466.507** |
| P_potts_6 | 50k/300181 | 17609.800 | 17362.600 | 17635.791 | **16756.999** |
| P_potts_7 | 50k/299735 | **16962.500** | **16962.500** | 19532.817 | 17002.578 |
| P_potts_8 | 50k/374169 | **24552.400** | 24596.800 | 25446.235 | **24552.413** |
| P_potts_9 | 50k/375603 | 25099.800 | 25095.600 | 25502.495 | **25050.522** |
| P_random_1 | 1k/7540 | **-4901.100** | -4505.020 | -4900.759 | -4564.763 |
| P_random_2 | 5k/37488 | -24059.900 | -22934.000 | **-24139.194** | -21834.693 |
| P_random_3 | 10k/74518 | -47873.200 | -47002.000 | **-48107.172** | -42120.325 |
| P_random_4 | 50k/249554 | 12881.500 | 14342.300 | 12233.890 | **11769.934** |
| P_random_5 | 50k/249374 | 12478.300 | 13337.000 | 12835.994 | **11750.969** |
| P_random_6 | 50k/299601 | 16723.600 | 16754.500 | 18031.964 | **16700.674** |
| P_random_7 | 50k/299538 | **16689.200** | 16701.600 | 18179.548 | **16689.252** |
| P_random_8 | 50k/374203 | **24556.000** | **24556.000** | 25549.594 | **24555.995** |
| P_random_9 | 50k/374959 | **24635.600** | 24689.500 | 25908.500 | **24640.039** |

Table 2: Results on the synthetic **high order** MRFs. Numbers correspond to the obtained energy values. Best in bold. "NA" denotes that no solution was found within the specified time limits. Best in bold.

| Graph | #Nodes/#cliques | Toulbar2 | NEUROLIFTING |
|---|---|---|---|
| H_Instances_1 | 500/12809 | -29359.827 | **-29835.757** |
| H_Instances_2 | 500/57934 | NA | **-20300.795** |
| H_Instances_3 | 50k/104059 | 1423.823 | **-3601.724** |
| H_Instances_4 | 50k/279293 | 10747.544 | **9782.693** |
| H_Instances_5 | 50k/229727 | 10534.909 | **9371.913** |

**Pairwise cases.** We evaluate pairwise cases from the UAI MPE dataset. The full results of NEUROLIFTING are detailed in Appendix B. From Table 3, we see that on trivial pairwise cases, where Toulbar2 successfully identifies the optimal solutions, NEUROLIFTING achieves comparably high-quality solutions that are on par with those obtained by LBP and TRBP. In cases where the problems become more challenging, although NEUROLIFTING does not surpass Toulbar2, it outperforms both LBP and TRBP. This suggests that NEUROLIFTING demonstrates improved performance on real-world datasets compared to simpler artificial instances.

**High-order cases.** For the high-order cases, we select a subset that has relatively large sizes. The results are presented in Table 4. The performance of NEUROLIFTING aligns with the results obtained from synthetic instances, demonstrating superior efficacy on larger problems while consistently outperforming Toulbar2 in dense cases.

## 4.3 PHYSICAL CELL IDENTITY

Physical Cell Identity (PCI) is an important parameter used in both LTE (Long-Term Evolution) and 5G (Fifth Generation) cellular networks. It is a unique identifier assigned to each cell within the network to differentiate and distinguish between neighboring cells. We transform PCI instances into pairwise MRFs, thus all the baselines could be evaluated. Appendix F details how to perform the transformation.

We employ an internal real-world PCI data collection along with a synthetic PCI dataset for evaluation. The configurations for LBP, TRBP, and our proposed NEUROLIFTING approach are consistent with those outlined in Section 4.1. For the Toulbar2 method, a time limit of 3600 seconds is set, while other parameters remain at their default values. The results are summarized in Table 5. The first five instances are real-world PCI cases sourced from a city in China, while the latter five instances are generated. We see for smaller problem instances, Toulbar2 is able to solve them exactly.

Table 3: Results on the UAI inference competition 2022. Numbers correspond to the obtained energy values. Best in bold."opt" denotes it is the optimal solution.

| Graph | #Nodes/#Edges | LBP | TRBP | Toulbar2 (1200s) | NEUROLIFTING |
|---|---|---|---|---|---|
| ProteinFolding_11 | 400/7160 | -3106.080 | 3079.030 | **-4461.047** | -4065.294 |
| ProteinFolding_12 | 250/1848 | 3570.210 | 3604.240 | **3562.387(opt)** | 16051.798 |
| Grids_19 | 1600/3200 | -2250.440 | -2103.610 | **-2643.107** | -2404.975 |
| Grids_21 | 1600/3200 | -13119.300 | -12523.300 | **-18895.393** | -16446.410 |
| Grids_24 | 1600/3120 | -13210.400 | -13260.900 | **-18274.302** | -16008.008 |
| Grids_25 | 1600/3120 | -2170.890 | -2171.050 | **-2620.268** | -2353.223 |
| Grids_26 | 400/800 | -2063.350 | -1903.910 | **-3010.719** | -2608.395 |
| Grids_27 | 1600/3120 | -9024.640 | -9019.470 | **-12284.284** | -10704.057 |
| Grids_30 | 400/760 | -2142.890 | -2154.910 | **-2984.248** | -2691.091 |
| Segmentation_11 | 228/624 | 329.950 | 339.762 | **312.760 (opt)** | 334.882 |
| Segmentation_12 | 231/625 | 75.867 | 77.898 | **51.151 (opt)** | 79.151 |
| Segmentation_13 | 225/607 | 75.299 | 88.554 | **49.859 (opt)** | 69.430 |
| Segmentation_14 | 231/632 | 95.619 | 98.691 | **92.334 (opt)** | 94.951 |
| Segmentation_15 | 229/622 | 412.990 | 418.853 | **380.393 (opt)** | 386.701 |
| Segmentation_16 | 228/610 | 100.853 | 101.670 | **95.000 (opt)** | 98.209 |
| Segmentation_17 | 225/612 | 421.888 | 432.012 | **407.065 (opt)** | 425.240 |
| Segmentation_18 | 235/647 | 100.389 | 98.411 | **82.669 (opt)** | 88.809 |
| Segmentation_19 | 228/624 | 86.589 | 86.692 | **58.704 (opt)** | 70.770 |
| Segmentation_20 | 232/635 | 289.435 | 291.527 | **262.216 (opt)** | 298.802 |

Table 4: Results on high-order cases of the UAI inference competition 2022. Numbers correspond to the obtained energy values. Best in bold.

| Graph | #Nodes/#cliques | Toulbar2 (1200s) | Toulbar2 (3600s) | NEUROLIFTING |
|---|---|---|---|---|
| Maxsat_gss-25-s100 | 31931/96111 | **-145969.060** | -145969.060 | -143158.612 |
| BN-nd-250-5-10 | 250/250 | 155.129 | **154.610** | 180.917 |
| Maxsat_mod4block_2vars_10gates_u2_autoenc | 479/123509 | -186103.111 | -186103.111 | **-187416.656** |
| Maxsat_mod2c-rand3bip-sat-240-3.shuffled-as.sat05-2520 | 339/2416 | -3734.627 | **-3737.076** | -3732.294 |
| Maxsat_mod2c-rand3bip-sat-250-3.shuffled-as.sat05-2535 | 352/2492 | **-3863.259** | -3863.259 | -3852.584 |

However, as the problem scale increases, it becomes increasingly challenging for Toulbar2 to effectively explore the solution space, and both LBP and TRBP struggle to converge. In contrast, NEUROLIFTING demonstrates strong generalization ability across all scales. Notably, it achieves commendable performance on large scales.

## 4.4 ANALYSIS AND ABLATION STUDY

**Choice of GNN backbones.** We evaluate the model's performance when implemented with different GNN backbones, as classified in Section 3.3. We compare their average performance across several datasets: pairwise cases from the UAI Inference Competition 2022, real-world PCI instances from our private dataset, and synthetic instances that we generated. Each synthetic instance comprises 1000 nodes with an average degree of either 4 or 8. The cases studied include both random energy configurations and Potts energy models, allowing a comprehensive assessment. From Fig. 3, we observe that across all datasets, GraphSAGE achieves the best results and exhibits the fastest convergence.

Table 5: Results on the PCI instances. Numbers are the obtained energy values. Best in bold.

| Graph | #Nodes/#cliques | LBP | TRBP | Toulbar2 (3600s) | NEUROLIFTING |
|---|---|---|---|---|---|
| PCI_1 | 30/165 | 20.344 | 20.455 | **18.134** | 18.718 |
| PCI_2 | 40/311 | **98.364** | 98.762 | **98.364** | 100.662 |
| PCI_3 | 80/1522 | **1003.640** | **1003.640** | **1003.640** | 1009.202 |
| PCI_4 | 286/10714 | 585.977 | 585.977 | 426.806 | **415.677** |
| PCI_5 | 929/29009 | 1591.590 | 1591.590 | 1118.097 | **1087.291** |
| PCI_syththetic_1 | 280/9678 | 564198.000 | 568082.000 | 522857.923 | **496685.831** |
| PCI_syththetic_2 | 526/34500 | 2.092e+06 | 2.084e+06 | 2.064e+06 | **1.907e+06** |
| PCI_syththetic_3 | 1000/49950 | 2.932e+06 | 2.908e+06 | 2.856e+06 | **2.672e+06** |
| PCI_syththetic_4 | 1500/78770 | 4.568e+06 | 4.532e+06 | 4.534e+06 | **4.186e+06** |
| PCI_syththetic_5 | 2000/120024 | 6.807e+06 | 6.904e+06 | 7.023e+06 | **6.520e+06** |

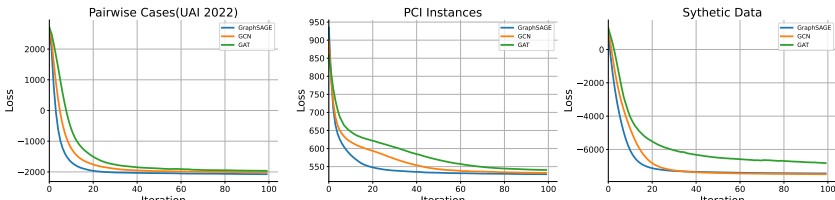

Figure 3: The average loss curves over UAI incference competition 2022 pairwise cases, PCI instances and systhetic instances using GraphSAGE, GCN and GAT as the GNN backbones.

**Choice of Optimizer.** The selection of the optimizer is discussed in Section 3.4, based on an analysis of the problem structure and empirical trials. We evaluate three optimizers: SGD, RMSprop, and Adam, using pairwise cases from the UAI 2022 dataset. The learning rate is set to $10^{-4}$, the embedded feature vector dimension is 1024, and we employ an 8-layer network. These configurations are consistent across all test cases for each optimizer. Results with average loss curves

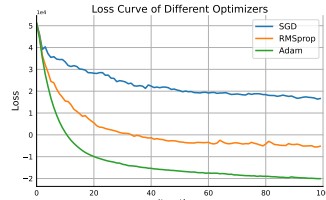

in the right figure, illustrating the differences in convergence rates and final results. We see that Adam outperforms both RMSprop and SGD in terms of convergence speed and stability.

**Loss Landscape Visualization.** We utilize the tool developed by Li et al. (2018) to visualize the loss landscape. Detailed settings of the visualization is in Appendix D. We visualize the evolution of the loss landscape for networks with varying depths, specifically for $K \in \{1, 2, 5, 8\}$. The resulting landscape visualizations are presented in Fig. 4, as well as the converged loss change trend in Fig. 5. We observe that a significant portion of the loss function is relatively flat, indicating that the loss can only decrease in constrained regions of the parameter space. As more layers are incorporated into the lifted model, it effectively expands these local regions, facilitating convergence toward better solutions. This characteristic suggests that the lifted model provides a greater capacity to navigate the optimization landscape.

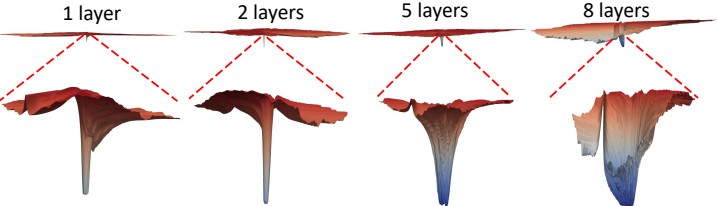
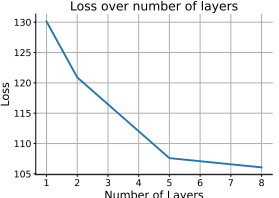

Figure 4: The landscape of instance Segmentation_19. From top to the bottom, each column correspond to network layer $\{1, 2, 5, 8\}$. The first row is the landscape range from $[-10, +10]$ for both $\delta$ and $\eta$ direction. The second row is the landscape range from $[-1, +1]$ for both $\delta$ and $\eta$ direction.

Figure 5: The training loss of instance Segmentation_19 after convergence of using network layer number $\{1, 2, 5, 8\}$.

**Related work** and **More analysis** are in Appendix A and C, respectively.

## 5 CONCLUSION

In this paper, we introduced NEUROLIFTING and its application to solving MAP problems for MRFs. Our experiments showed that NEUROLIFTING effectively handles MRFs of varying orders and energy functions, achieving performance on par with established benchmarks, as verified on the UAI 2022 inference competition dataset. Notably, NEUROLIFTING excels with large and dense MRFs, outperforming traditional methods and competing approaches on both synthetic large instances and real-world PCI instances. This method, which utilizes Neural Networks for lifting, has proven successful and could potentially be extended to other optimization problems with similar modeling frameworks.

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

## A    RELATED WORK

**MRF and Inference.** In Markov Random Fields (MRFs), the energy function is associated with a graph-structured probability distribution. A key inference challenge is determining the maximum a posteriori (MAP) configuration. Although minimizing the energy function is NP-hard, advances in inference techniques have enhanced model capabilities. For cycle-free graphs, the MAP problem can be effectively addressed using a variant of the min-sum algorithm (Clifford & Hammersley, 1971; Besag, 1974; Kumar et al., 2005) , which extends the Viterbi algorithm (Yedidia et al., 2003) to arbitrary cycle-free structures. In graphs with cycles, graph cut methods (Komodakis et al., 2007; Roy & Cox, 1998; Boykov et al., 1998; Ishikawa & Geiger, 1998; Szummer et al., 2008) utilize min-cut/max-flow strategies to efficiently minimize energy, although they require MRFs to be graph-representable and are unsuitable for multi-labeled MRFs. Two graph-cut-based strategies (Ishikawa, 2003; Schlesinger & FLACH, 2006) have been developed: the label-reduction method, for specific MRFs requiring binary conversion, and the move-making method, influenced by the size of node state combinations.

The belief propagation (BP) algorithm (Pearl, 1982; 1988) , introduced by Pearl in 1982, is a widely used iterative inference method for Bayesian networks, functioning through message passing. However, BP struggles with loops, leading to loopy belief propagation (LBP) (Weiss & Freeman, 2001; Felzenszwalb & Huttenlocher, 2004; Frey & Mackay, 2002) , which iterates message passing even in cyclic graphs. While LBP has shown effectiveness in vision tasks, it lacks guaranteed convergence. Recent advancements aim to enhance BP's performance, such as adjusting message significance with discount factors (Grim & Felzenszwalb, 2023) and constructing hierarchical frameworks for large-scale MRFs (Yan et al., 2023). The Junction Tree Algorithm (JTA) (Aji & McEliece, 2000) provides exact inference for arbitrary graphs but is NP-hard, limiting its practicality. In pairwise MRFs, integer linear programming (ILP) formulations yield solutions through tree-reweighted message passing (TRBP) (Wainwright et al., 2005) , which includes edge-based and tree-based schemes, though they lack guaranteed convergence. The sequential TRW-S (Kolmogorov, 2006) scheme achieves weak tree agreement, ensuring lower bounds stabilize, but requires substantial time for convergence.

**Lifting in Optimization.** Lifting techniques have garnered significant attention in the optimization field, particularly in tackling combinatorial problems and enhancing the performance of various algorithms (Marchand et al., 2002). These techniques involve transforming a problem into a higher-dimensional space, which facilitates more effective representation and solution strategies. They are applied to both mixed 0-1 integer programming problems (Balas et al., 1993) and more general mixed-integer programming (MIP) problems in conjunction with primal cutting-plane algorithms (Dey & Richard, 2008). Additionally, lifting techniques have been integrated with variable upper bound constraints in applications such as the Knapsack problem (Shebalov et al., 2015). The use of lifting methods has also extended into robust optimization scenarios (Georghiou et al., 2020; Bertsimas et al., 2019). Furthermore, combining lifting techniques with Newton's method has shown promise in addressing nonlinear optimization problems (NLPs) (Albersmeyer & Diehl, 2010).

**Unsupervised GNNs for Combinatorial Optimization.** Graph Neural Networks (GNNs) have been proved to be powerful in optimization (Yu et al., 2019; Ying et al., 2024) and recent advancements in unsupervised GNNs have demonstrated their effectiveness in tackling combinatorial optimization problems. By leveraging the structural properties of graph data, unsupervised GNNs can learn meaningful representations of nodes and edges without requiring labeled datasets. It was shown that unsupervised GNNs can effectively capture the combinatorial structure inherent in these problems, leading to improved heuristics and solution strategies  (Peng et al., 2021). This capability is particularly advantageous for problems such as the Traveling Salesman Problem (TSP) (Gaile et al., 2022; Min et al., 2023), the Vehicle Routing Problem (VRP) (Wu et al., 2024) and Boolean satisfiability problem(SAT)  (Cappart et al., 2023), where traditional optimization methods often face challenges related to scalability and solution quality. The Max Independent Set (MIS) and Max Cut problems can also be solved efficiently in this way (Schuetz et al., 2022). However, the loss functions may lack the flexibility to effectively handle higher-order relationships beyond mere edges.

Table 6: Results on the UAI inference competition 2022 of NEUROLIFTING with different feature dimensions. Numbers correspond to the obtained energy values.

| Graph | #Nodes/#Edges | dim=64 | dim=512 | dim=1024 | dim=4096 | dim=8192 |
|---|---|---|---|---|---|---|
| ProteinFolding_11 | 400/7160 | -3892.949 | -3886.701 | -3946.168 | 4065.294 | -4003.323 |
| ProteinFolding_12 | 250/1848 | 16064.795 | 16068.406 | 16051.798 | 16088.073 | 16071.324 |
| Grids_19 | 1600/3200 | -2355.159 | -2404.975 | -2337.281 | -2341.2746 | -2373.618 |
| Grids_21 | 1600/3200 | -16478.466 | -16169.0320 | -16446.410 | -16209.017 | -16278.668 |
| Grids_24 | 1600/3120 | -16008.008 | -15900.249 | -15841.799 | - 15608.162 | -15948.219 |
| Grids_25 | 1600/3120 | -2343.547 | -2353.223 | -2319.899 | -2306.686 | -2288.182 |
| Grids_26 | 400/800 | -2532.837 | -2608.395 | -2553.781 | -2559.572 | -2535.464 |
| Grids_27 | 1600/3120 | -10748.024 | -10704.057 | -10514.857 | -10389.031 | -10665.737 |
| Grids_30 | 400/760 | -2563.274 | -2631.862 | -2640.044 | -2691.091 | -2649.462 |
| Segmentation_11 | 228/624 | 330.541 | 349.906 | 334.882 | 356.895 | 337.312 |
| Segmentation_12 | 231/625 | 74.705 | 74.029 | 155.062 | 79.151 | 105.801 |
| Segmentation_13 | 225/607 | 67.371 | 86.064 | 69.430 | 72.394 | 112.516 |
| Segmentation_14 | 231/632 | 94.192 | 96.501 | 100.582 | 104.091 | 96.572 |
| Segmentation_15 | 229/622 | 388.223 | 386.701 | 397.246 | 407.731 | 390.641 |
| Segmentation_16 | 228/610 | 99.086 | 99.690 | 111.121 | 98.209 | 108.360 |
| Segmentation_17 | 225/612 | 424.686 | 426.130 | 425.192 | 425.240 | 427.810 |
| Segmentation_18 | 235/647 | 89.905 | 101.307 | 94.224 | 88.854 | 88.809 |
| Segmentation_19 | 228/624 | 76.244 | 78.337 | 74.284 | 69.116 | 70.770 |
| Segmentation_20 | 232/635 | 298.802 | 301.802 | 302.673 | 304.457 | 312.970 |

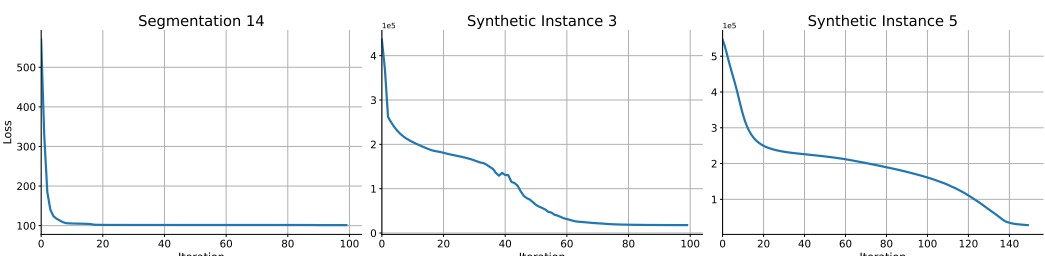

Figure 6: The loss curves of the Segmentation_14, P_potts_6 and P_potts_8 from pairwise potts synthetic problems.

## B FULL TABLE OF UAI PAIRWISE CASES

In Table 6, we present the inference results of NEUROLIFTING using various dimensions of feature embeddings applied to the pairwise cases from the UAI Inference Competition 2022. The results indicate that the dimensionality of the feature embeddings is indeed a factor that influences model performance. However, in most cases, a moderate dimension is sufficient to achieve high-quality results. This suggests that while increasing dimensionality may provide some advantages, the decision should be made by considering both performance and computational efficiency.

## C MORE ANALYSIS

**Efficiency vs Solution Quality.** We evaluate the performance of the NEUROLIFTING using the same network size and a consistent learning rate of 1e-4 on the Segmentation_14 dataset from the UAI 2022 inference competition, along with two of our generated Potts instances: P_potts_6 and P_potts_8. This setup allows us to observe the trends associated with changes in graph size and sparsity. The results are presented in Fig. 6. It is seen that the model converges rapidly when the graph is small and sparse, within approximately 20 iterations on the Segmentation_14 dataset. Comparing P_potts_6 and P_potts_8, we observe that though both graphs are of the same size, the denser graph raises significantly more challenges during optimization. This indicates that increased size and density can complicate the optimization process, and NEUROLIFTING would need more time to navigate a high quality solution under such cases.

## D  Visualization setup

The core idea of the visualization technique proposed by Li et al. (2018) involves applying perturbations to the trained network parameters $\theta^*$ along two directional vectors, $\delta$ and $\eta$: $f(\alpha, \beta) = L(\theta^* + \alpha\delta + \beta\eta)$. By doing so, we can generate a 3-D representation of the landscape corresponding to the perturbed parameter space. We sampled 250000 points in the $\alpha - \beta$ plane, where both $\alpha$ and $\beta$ range from -10 to 10, to obtain an overview of the loss function landscape. Subsequently, we focused on the region around the parameter $\theta^*$ by sampling an additional 10,000 points in a narrower range, with $\alpha$ and $\beta$ both from $-1$ to 1.

## E  Read UAI format files

An example data file in UAI format is provided in Box E. This Markov Random Field consists of 3 variables, each with 2 possible states. Detailed information can be found in the box, where we illustrate the meanings of different sections of the file. Notably, in the potential section, the distributions are not normalized. During the belief propagation (BP) procedure, these distributions will be normalized to prevent numerical issues. However, in the energy transformation phase, we will utilize these values directly.

---

**Example.uai**

```
MARKOV      //Instance type
3     //Number of vairables
2 2 2     //State number of each variable
5     //Number of cliques that has potentials
1 0     //1 means this clique is a variable, and the variable is 0.
1 1
1 2
2 0 1     //2 means this clique is an edge, the edge is (0, 1).
3 0 1 2     //3 means this clique includes 3 variables, and the clique is (0, 1, 2).

2     //The number 2 indicates that the potential in the next line has two values.
0.1 0.9     //The potential of variable 0 is 0.1 for state 0 and 0.9 for state 1.

2
0.1 10

2
0.5 0.5

4
0.1 1.0 1.0 0.1//The potential of the state combinations for variables 0 and 1 is given in the
order of (0,0), (0,1), (1,0) and (1,1).

8
0.1 2.0 0.1 0.1 0.1 0.1 0.1 2.0     //The potential of the state combinations for variables 0, 1,
and 2 is given in the order of (0,0,0), (0,0,1), (0,1,0), (0,1,1), (1,0,0), and so on.
```

---

Since the transformation of variable energies and clique energies follows the same procedure, we will use the edge $(0, 1)$ to illustrate the transformation. The value calculations will adhere to Equation 1. In Table 7, we present the unnormalized joint distribution for the edge $(0, 1)$, while Table 8 displays the energy table for the edge $(0, 1)$ after transformation.

## F  PCI problem formulation

The Mixed Integer Programming format of PCI problems is as follows:

Table 7: $P(x_0, x_1)$

| $x_0$ \ $x_1$ | 0 | 1 |
|---|---|---|
| 0 | 0.1 | 1.0 |
| 1 | 1.0 | 0.1 |

Table 8: $\theta_C(x_0, x_1)$

| $x_0$ \ $x_1$ | 0 | 1 |
|---|---|---|
| 0 | 2.303 | 0 |
| 1 | 0 | 2.303 |

$$\min_{z,L} \quad \sum_{(i,j)\in\mathcal{E}} a_{ij}L_{ij} \tag{8}$$

$$\text{s.t.} \quad z_{np} \in \{0,1\}, \quad \forall n \in N, p \in P \tag{8a}$$

$$\sum_{p\in P} z_{np} = 1, \quad \forall n \in N. \tag{8b}$$

$$\sum_{p\in M_{ih}} z_{n_i p} + \sum_{p\in M_{jh}} z_{n_j p} - 1 \le L_{ij}, \forall (i,j) \in \mathcal{E}, \forall h \in \{0,1,2\}. \tag{8c}$$

where $n$ is the index for devices, and $N$ is the set of these indices. $P$ stands for the possible states of each device. $M_{ih}$ stands for the possible states set for node $n_i$. $L_{ij}$ is the cost when given a certain choices of the states of device $i$ and device $j$, $a_{ij}$ is the coefficient of the cost in the objective function. There is an $(i,j) \in \mathcal{E}$ means there exists interference between these two devices. When using MRF to model PCI problems, each random variable represent the identity state of the given node and the interference between devices would be captured by the pairwise energy functions. Next we will introduce how to transform the PCI problem from MIP form to MRF form.

In the original MIP formulation of the PCI problems, three types of constraints are defined. By combining Equation 8a and Equation 8b, we establish that each device must select exactly one state at any given time. Furthermore, the constraint in Equation 8c indicates that interference occurs between two devices only when they select specific states. The overall impact on the system is governed by the value of $L_{ij}$ and its corresponding coefficient. Given that interference is always present, the objective is to minimize its extent.

To transform these problems into an MRF framework, we utilize Equation 8b to represent the nodes, where each instance of Equation 8a corresponds to the discrete states of a specific node. The constraints set forth in Equation 8a and Equation 8b ensure that only one state can be selected at any given time, thus satisfying those conditions automatically. By processing Equation 8c, we can identify the edges and their associated energies. If $z_{n_i p}$ and $z_{n_j p}$ appear in the same constraint from Equation 8c, we can formulate an edge $(i,j)$. By selecting different values for $z_{n_i p}$ and $z_{n_j p}$, we can determine the minimum value of $L_{ij}$ that maintains the validity of the constraint.

The product of $L_{ij}$ and $a_{ij}$ represents the energy associated with the edge $(i,j)$ under the combination of the respective states. Once the states of all nodes are fixed, the values of the edge costs also become fixed. This leads to the conclusion that the objective function is the summation of the energies across all edges. Since the PCI problems do not include unary terms, we can omit them during the transformation process. This establishes a clear pathway for converting the MIP formulation into an MRF representation, allowing us to leverage MRF methods for solving the PCI problems effectively.

**Example**
The original problem is

$$\min_{z,L} \quad L_{1,2} + 3L_{2,3}$$

$$\text{s.t.} \quad z_{np} \in \{0,1\}, \qquad\qquad\qquad \forall n \in \{1,2,3\}, p \in \{1,2,3\}$$

$$\sum_{p \in P} z_{np} = 1, \qquad\qquad\qquad\qquad \forall n \in \{1,2,3\}.$$

$$z_{11} + z_{21} - 1 \le L_{1,2}$$
$$z_{13} + z_{22} - 1 \le L_{1,2}$$
$$z_{12} + z_{23} - 1 \le L_{1,2}$$
$$z_{21} + z_{31} - 1 \le L_{2,3}$$
$$z_{22} + z_{32} - 1 \le L_{2,3}$$
$$z_{23} + z_{33} - 1 \le L_{2,3}$$

$$(9)$$

Then the corresponding MRF problem is

$$\min \theta_{1,2}(x_1, x_2) + \theta_{2,3}(x_2, x_3) \tag{10}$$

the energy on edge $(x_1, x_2)$ and edge $(x_2, x_3)$ are as follows:

| $x_1$ \ $x_2$ | $z_{21}$ | $z_{22}$ | $z_{23}$ |
|---|---|---|---|
| $z_{11}$ | 1 | 0 | 0 |
| $z_{12}$ | 0 | 0 | 1 |
| $z_{13}$ | 0 | 1 | 0 |

| $x_2$ \ $x_3$ | $z_{31}$ | $z_{32}$ | $z_{33}$ |
|---|---|---|---|
| $z_{21}$ | 3 | 0 | 0 |
| $z_{22}$ | 0 | 3 | 0 |
| $z_{23}$ | 0 | 0 | 3 |