# OpenReview forum: "NeuroLifting: Neural Inference on Markov Random Fields at Scale"
_ICLR.cc/2025/Conference — Submitted to ICLR 2025_

### Official Review · Reviewer_qPjP · 2024-10-25

**Soundness:** 1
**Presentation:** 1
**Contribution:** 2
**Rating:** 5
**Confidence:** 3

**Summary:**

The paper investigates approximating Markov random fields (MRF) by optimizing a graph neural network (GNN) to maximize a surrogate to its joint density.
It suggests using learnable padded embeddings of the MRF decision variables as GNN features, allowing gradient-based optimization.
The paper claims that:
 1. It is the first to introduce neural network-based lifting to MRF inference.
 2. The approach, called NeuroLifting, simplifies the optimization process for large MRFs.
 3. Loss computation with NeuroLifing scales linearly with the MRF nodes and enables parallelization and GPUs accelerated computation.
 4. It empirically demonstrates that NeuraLifing finds near-optimal solutions on moderate-size MRFs and solutions superior to baselines on large MRFs.

**Strengths:**

- The central idea of embedding decision variables in an MRF to use a GNN is original.
- The selection of GraphSAGE for convolution is well justified and supported by empirical results (Figure 3).

**Weaknesses:**

## Reasons for score
- The abstract states that NeuroLifing "delivers superior solution quality against all baselines" on large-scale MRFs. However, Table 1 does not show this, and two baselines (LBP and TRBP) need to be included in Table 2. Please consider modifying the abstract to reflect your results better. For example, you could restrict your claim to Toulbar2, i.e., exact methods.
- The experimental results lack standard error estimation, which makes method comparison difficult because it's hard to judge the significance of the difference in energy. Considering your method is probabilistic, you could use multiple initializations to estimate the standard error.
- A convergence criterion needs to be clearly stated for NeuroLifting UAI22 dataset.
- Related work is delegated to the appendix.
- It needs to be clarified whether NeuroLifing is guaranteed to produce a valid solution. In particular,
  - Please explain how the padding scheme suggested produces only valid solutions, but padding with the largest of all energies can produce infeasible solutions.
  - Please explain with evidence (either theoretical or empirical) how the rounding scheme for $p_i(\theta)$ avoids multi-assignment issues.

**Questions:**

1. Does a GNN with jumping knowledge preserve the Markov assumption?
  2. Why does Table 2 not include LBP and TRBP?
  3. Was a hard time constraint of 1200 seconds enforced for LBP, TRBP and NeuroLifing?
  4. Is NeuroLifing sensitive to the initialization of the embeddings and network?
  5. Why does Toulbar2 fail on H_Instances_2?
  6. Table 1 reports "the final loss given by the loss function," whereas the other tables report the final energy. It needs to be clarified that the distinction between loss and energy exists for the baseline methods. If so, what is the gap between loss and energy?
  7. What is the gap between loss and energy for NeuroLifting on synthetic, UAI and PCI?
  8. The interpretation of the linear complexity needs to be clarified. By mentioning the linear complexity in node size, is the paper claiming an asymptotic improvement over alternatives?
  9. How does the suggested padding scheme produce only valid solutions, whereas padding with the largest of all energies can produce infeasible solutions?
  10. How does the rouding scheme for $p_i(\theta)$ avoid multi-assignment issues?
  11. What does "showing remarkable scalability and efficiency" mean in (lines 71-72)? What is the concrete evidence that shows this claim?
  12. What stopping criteria are used for NeuroLifting on UAI?

##  Minor comments (did not affect score)
  1. Tables 1, 2, 3 and 4 would be more legible and convey the same results if reported as whole numbers instead.
  2. Capitalize the titles of the references.
  3. Use only one of: `DOI:`, `ISBN` or neither for book references.
  4. The caption for Figure 1 should describe the stages in the diagram. Figure 2 does this well.
  5. In Figure 1 $\mathcal{P}$ and $H$ are not defined.
  6. Cross entropy is not defined.
  7. In eq. 5, $v_i(\theta)$ is a mapping, but above $v_i$ is defined as a bit string (not a mapping).
  8. Cliques should be capitalized in Tables 1, 2, 4 and 5.
  9. $\mathcal{X}$ is not clearly defined. From the padding paragraph and section 3.4, it looks like it should be $x_i \in \mathcal{X}_i$ and $\mathcal{X} = \cup_i \mathcal{X}_i$.
  10. Table 5 should use scientific notation.
  11. Is the learning rate really $1e^{-4} \approx 0.018$ in 4.1?

---

> ### Author Response · Authors · 2024-11-15
>
> W stands for weakness, Q stands for question.
>
> **W1/Q2**: We didn't provide the result of LBP and TRBP since it is hard for them to deal with high order MRFs with this magnitude. They reported error message even for the toy examples. That is why we have no test on the hard problems that listed in Tabel 2 and Table 3. You can see the result dsiplay on the website of OpenGM (http://old.ilastik.org/opengm/index.php?l0=benchmark)(the source of LBP and TRBP) , they only test these two methods on very simple high order MRFs.
>
> **W2**: We have listed these statistics in the general response.
>
> **W3**:  The time limit is set to 1200 seconds, consistent with the settings used in Toulbar2. We also employ 100 iterations, and the stopping criteria remain the same as those in our synthetic problems, which include an absolute tolerance of 1e-4 and a patience parameter of 10. I apologize for any ambiguity in the previous presentation.
>
> **W4**: We made this decision due to the page limit imposed at this stage, as we wanted to present as much content as possible in our paper. We will reorganize the structure in the official version to enhance clarity and presentation.
>
> **W5**: See the answers for Q9 and Q10.
>
> **Q1**: The tricks of GNNs won't change the properties of the original model since we have already transferred from the original problem to the optimization problem on the GNN.
>
> **Q3**: Yes, the programme will terminate if the time exceeds the timelimit for all these methods.
>
> **Q4**: The effects of different network structures are evident in the ablation study. The primary difference observed relates to convergence speed rather than the final results. We tested various initialization methods for the GNN, including uniform, Kaiming, and Xavier, but the differences between Xavier and Kaiming were minor(see Table 6 in the general response), which is why we did not include these results in the ablation study. Additionally, the initialization of the embeddings is also not particularly sensitive, as the results across the five trials in our experiments show minimal variation in most cases.**
>
> **Q5**: The results presented in Table 2 indicate that as the graph density increases, it becomes more challenging for Toulbar2 to identify a feasible solution. This behavior is expected, given that Toulbar2 employs a branch-and-cut algorithm, and the difficulty may be linked to the specific techniques they apply to graphical models; however, we are not privy to their implementation details.
>
> **Q6**: It is a typo here. The gap between these two terms is close given the experiment results, since after convergence, the GNN will assign very small probabilities to other labels. The contribution of these parts is minor. You can refer to the data we provide in the next question.
>
> **Q7** : We have listed all the losses and energies in tables in general response.
>
> **Q8**: The computation of our model is straightforward. It consists of the time taken for the GNN to process the graph and the time required for loss calculation. For the GNN component, the time complexity is linear with respect to the number of nodes, which is evident. Similarly, the loss calculation involves simple tensor multiplication, resulting in a linear time complexity in relation to the number of items.
>
> We are primarily analyzing the time complexity of our model. Notably, the time complexity for the LBP is also linear with respect to the node size, while the time complexity for the solver remains unknown."
>
> **Q9**：In our experiments, we explored multiple padding strategies, including the approach of padding with infinities. While we discovered that our current padding strategy emerged as the most effective solution based on our evaluations. We have provided an explanation of our chosen method in the Remarks section of 3.2.
>
> Additionally, we would like to propose another possible issue associated with padding using infinities. Specifically, when infinities are employed as padding values, even minor fluctuations in the neural network's output can lead to significant variations in the loss function. This sensitivity can complicate the optimization process, making convergence more difficult.
>
> **Q10**: In our scenario, a situation may arise where the selection of different labels results in the same energy, indicating that the choice of label becomes irrelevant; we could simply choose one at random. However, this occurrence is not observed during our process, suggesting that it is uncommon. Consequently, the rounding scheme we employed is sufficient to select a valid configuration.
>
> **Q11**: It is straightforward to see the scalability of NeuroLifting from the PCI instances and the synthetic problems. We get remarkable results for the cases from a few hundred nodes to 50k nodes and we test different densities for each of these cases under the same network settings.  For efficiency part, each of these test has the same time limits as the competitors.

---

> ### Author Response · Authors · 2024-11-15
>
> **Q12**：The time limit is set to 1200 seconds, consistent with the settings used in Toulbar2. We also employ 100 iterations, and the stopping criteria remain the same as those in our synthetic problems, which include an absolute tolerance of 1e-4 and a patience parameter of 10. We apologize for any ambiguity in the previous presentation.
>
> **MInor comments**:
> Thank you for the reminders; we will correct them in the final version of our paper. The learning rate of 1×10−4 is denoted as 1e-4.

---

> ### Comment · Reviewer_qPjP · 2024-11-19
> **Responds 1 (NeuroLifting)**
>
> Thank you for your rebuttal. You have adequately addressed weaknesses W2 through W4, so I have adjusted my score to a 5. However, I still have questions regarding the remaining weaknesses, which I need you to address to increase your score further.
>
> ## W1/Q2,
> I appreciate why you cannot compare with LBP and TRBP in Table 2. Thanks for clarifying this. However, with the evaluation you've provided, W1 still stands. The claim that your method provides better solution quality against all baselines does not hold. For example, considering the error, Toulbar2 energy is well within the energy NeuroLifting achieves on H Instances 1 and 5. See the table below.
>
> | Graph         | Toulbar2 | NeuroLifting  |
> |---------------|----------|---------------|
> | H Instances 1 |  -29,359 | -29,854 ± 450 |
> | H Instances 2 |       NA |  20,286 ± 143 |
> | H Instances 3 |    1,423 |    -3,611 ± 9 |
> | H Instances 4 |   10,747 |    9,754 ± 27 |
> | H Instances 5 |   10,534 |  10,217 ± 848 |
>
> ## Q11
> Regarding scalability, you have a strong case for outperforming approximate and exact methods. This is evident from your ability to perform inferences on all the graphs, whereas the other methods cannot. However, this point has not been stated clearly in your responses or the paper itself.
>
> I can't entirely agree with your claim of achieving remarkable results. Your performance is worse than the baseline on smaller instances and only comparable to the exact method on 2 out of the 5 large graphs.
>
> When it comes to efficiency, time-constrained experiments are not suitable for supporting this claim. To support the claim, you should provide timings.
>
> ## Q9
> Could you provide a reference or more details to support the claim that: "Specifically, when infinities are employed as padding values, even minor fluctuations in the neural network's output can lead to significant variations in the loss function."?
>
> It is still unclear why your padding scheme is the only one that avoids pathological behavior. In Section 3.2, you mention that your scheme discourages selection from padded states by incentivizing the algorithm to choose the original, non-padded states with lower energies. You argue that the issue with alternative padding schemes is their tendency to dramatically alter the loss landscape, potentially leading to the selection of non-padded states. However, from my understanding, using a larger padding value should further incentivize the selection of low-energy, non-padded states.
>
> Could you clarify what this dramatical alteration that can happen is?

---

> > ### Author Response · Authors · 2024-11-20
> >
> > **W1/Q2**:We have aimed to cover a wide range of sizes with varying densities, including **pairwise MRFs** and **high-order MRFs** in the synthetic tests. Table 2, which you referenced, presents our results for high-order MRFs. H_instance_1 represents a smaller case with moderate density compared to H_instance_2, and we achieved better results in the worst case among the five trials.
> >
> > For H_instance_5, which is a larger case, we observed a higher standard error, indicating that in one or two trials we obtained worse results compared to Toulbar2. This is partly due to the GNN we employed, which relies on message passing along edges. Additionally, we need to address the cliques present in high-order MRFs, and we did not implement specific optimizations for these challenges. Consequently, the performance in this scenario is less stable. However, on average, our results are better, and instability only affects a few trials.
> >
> > **Q11**:Our goal is to address the significant and long-standing **large-scale** MRF inference challenge, as emphasized in the title of our paper "NeuroLifting: Neural Inference on Markov Random Fields **at Scale**". This has been demonstrated by the results from the synthetic problems (Tables 1 and 2), where we outperform Toulbar2 in most large cases. For PCI instances, we achieve better results in large cases for both real and synthetic problems. In smaller cases, which are easier to solve, we claim to attain results **comparable** to the baselines.
> >
> > For the clock time comparison, the time limits are set according to the same standards (1200 seconds and 3600 seconds) used in the UAI inference competition, which is widely accepted.
> >
> > To better understand the timing, we conducted a simple time comparison using the PCI instances utilized in Table 5 in our manuscript. Since the log file for Toulbar2 does not contain time information for each solution found during the search, we had to establish multiple time limits to facilitate the following comparisons. We recorded the results every 200 seconds. Results are summarized in the **Table 7** below. For the first three simple instances, Toulbar2 was able to solve to optimality in less than 1 second. The time used by NeuroLifting is indicated in parentheses after the energy values in **Table 7**. Our implementation of NeuroLifting is more efficient than Toulbar2 on the two larger instances. NeuroLifting achieved better solutions at 400 seconds for both the PCI_4 and PCI_5 instances. For PCI_4, NeuroLifting utilized all 100 iterations within 600 seconds, while Toulbar2 did not update its solution until it was forced to stop. It is important to note that our current implementation of the code is written in Python (for research purposes) and has not been fully optimized, leaving significant room for further improvements. With targeted optimizations, we anticipate even better efficiency and scalability in handling these instances.
> >
> > **Table 7** Time comparison on PCI instances
> > | Instances | Algorithm | 200s | 400s | 600s | 800s | 1000s | 1200s |
> > |------|-------------|----------------|----------------|----------------|----------------|-----------------|----------------|
> > |PCI_1| Toulbar2    | 18.134     |   |   |   |   |   |
> > |PCI_1| NeuroLifting |   18.732  (11s)   |   |   |   |   |   |
> > | | | | | | | | |
> > |PCI_2  | Toulbar2              |  98.364    |   |   |   |   |   |
> > |PCI_2      | NeuroLifting         |  100.460 (19s)      |   |   |   |   |   |
> > | | | | | | | | |
> > |PCI_3       | Toulbar2              | 1003.640     |   |   |   |   |   |
> > |PCI_3      | NeuroLifting         |  1009.247 (83s)   |   |   |   |   |   |
> > | | | | | | | | |
> > |PCI_4       | Toulbar2              |  428.299     | 426.806 | 426.806 | 426.806 | 426.806 | 426.806 |
> > |PCI_4      | NeuroLifting         |  431.918      | 425.444 |  422.896 |
> > | | | | | | | | |
> > |PCI_5       | Toulbar2              |  1128.244     | 1121.325 | 1121.325 |  1121.325 |  1121.325 | 1121.325|
> > |PCI_5      | NeuroLifting         |  1340.518     | 1177.226 |  1125.842 | 1112.025 | 1107.175 |  1104.623 |
> >
> > **Q9**: When we use large values for padding, we apply the same value to all energy terms. As a result, some nodes may select labels from the padded area. Due to the presence of large numbers, the loss calculation can result in excessive gradient updates during backpropagation, which may lead to an unstable training process for the model. This issue becomes more pronounced when the number of labels varies significantly between nodes.  One significant drawback of this approach is that the original maximum energy value in each term can vary widely. For instance, some may have a maximum of 1, while others could reach 10,000,000. Consequently, the padding values carry different meanings for each of these energies. Therefore, using an excessively large padding value may not be ideal for these scenarios.

---

> ### Comment · Reviewer_qPjP · 2024-11-25
> **Responds 2**
>
> First, I would like to thank the authors for the substantial additional evidence they have provided.
>
> ## W1
> "We achieved better results in the worst case among the five trials." This seems unusual, given that you report the standard error. However, the issue remains that the article asserts superior performance across the board without sufficient evidence to substantiate these claims. I recommend that the authors revise their manuscript to reflect their actual results more accurately, as this would significantly enhance the quality of the paper.
>
> ## Q11
> Thank you for sharing the timings. I would expect an efficient method to produce better results quicker than its alternatives. However, based on the experiment presented in Table 7, Toulbar2 consistently outperforms the NeuroLifting method at 200 seconds, indicating this expectation is not met. That said, I would agree with your perspective if efficiency were interpreted as the ability to integrate observations to achieve superior results.
>
> ## Q9
> Thanks for clarifying this point.
>
> With the authors' latest rebuttal, I have no further questions. I would like to keep my score as is.

---

> > ### Author Response · Authors · 2024-11-25
> >
> > We do get a better result on H_instance_1 even considering the standard error here. For the time comparison, 200s is not emperically feasible on lagre problems (due to insufficient convergence). This is also the reason why the widely accepted and challenging UAI protocol sets the default time horizon to be 1200s and 3600s. Under this fair comparison setting, we get a better solution with only a few more time on top of 200s. Considering real-world scenarios, we believe both the convergence speed and the performance is promising for large-scale MRF inference. Besides, we are compared with a fully developed c++ solver -- Toulbar2 -- with decades of effort. Therefore, we believe there is still huge potential to be improved on code level after the code is open-sourced to the community.
> >
> > Thanks for your valuable advice. We will make a concerted effort to improve our paper to better present our work.

---

### Official Review · Reviewer_HyCv · 2024-10-28

**Soundness:** 2
**Presentation:** 3
**Contribution:** 1
**Rating:** 3
**Confidence:** 2

**Summary:**

This paper focuses on the inference in large-scale Markov Random Fields (MRF), where conventional methods often suffer from either efficiency or solution accuracy. To address this issue, this paper advocates mapping the inference of MRF to a Graph Neural Networks (GNNs), which enables smooth loss landscape and scalable inference. Numerical experiments on several datasets showcases that the proposed NeuroLifting presents comparable performance to the non-scalable exact solver Toulbar2.

**Strengths:**

1. The organization of this paper is clear, and the writing is easy to understand.

2. The figures and the example provided in figure 1 are helpful for understanding the paper.

**Weaknesses:**

Several parts of the paper are confusing, as listed one-by-one below.

1. When introducing Lifting in Section 2, it is stated that "Lifting reformulates an optimization problem into a more tractable form by introducing auxiliary variables or constraints, making the optimal solutions more accessible." However in Section 3.3, the high-dimensional feature vectors are learnable and randomly initialized, without any guarantee on the equivalence between the original problem and the alternative GNN optimization problem. It would be beneficial if some theoretical analysis can be provided to ensure the GNN is actually/approximately solving the original problem, or an equivalent problem. In the current version, there is not even an objective function for the alternative optimization problem, making it hard to connect this paper with "Lifting".

2. This paper claims that NeuroLifting is a "non-parametric" method. Nevertheless, line 249 argued that "we optimize both the GNN parameters and those of the encoder," which conflicts with the proceeding claim.

3. Line 61 says that NeuroLifting "outperforms all existing approximate inference strategies in terms of solution quality without sacrificing computational efficiency." In the experiments, only the time limit for the exact solver Toulbar2 is reported, while it is unclear how NeuroLifting compares to conventional approaches in terms of both time and space complexities.

4. No codes or supplementary files are provided along with the submission. As a result, the reviewers cannot verify the results reported in the manuscript.

Since I'm not quite familiar with Lifting and MRFs, I will keep a low confidence score.

**Questions:**

See Weaknesses.

---

> ### Author Response · Authors · 2024-11-15
>
> W stands for weakness.
>
> **W1**: In this work, we do not transform the problem using traditional methods such as imposing relaxation or other transformations. Instead, we focus on transforming the original energy minimization problem by optimizing the neural network parameters to achieve the best mapping of node labels, thereby minimizing the loss. The lifting is accomplished through multiple layers of GNNs, which elevate the problem from node-level inference to the neural network parameters.
>
> Consequently, these two problems are not equivalent; we utilize the transformed problem to approximate the original one. In the paper by Yann Dauphin et al. (2014), titled 'Identifying and Attacking the Saddle Point Problem in High-Dimensional Non-Convex Optimization' (https://arxiv.org/abs/1406.2572), the authors discuss the prevalence of saddle points in the landscape of neural networks, leading to ease of optimization using gradients. This insight highlights why gradient-based optimization is more manageable, forming the foundation of our method, and also somewhat inspired us to solve complex MRFs using neural networks with gradient backpropagation.
>
> **W2**: The term 'non-parametric' does not imply the absence of tunable parameters within the model. Rather, it signifies that we do not require assumptions about the underlying population. Non-parametric methods do not depend on any specific parametric group of probability distributions and are often referred to as distribution-free tests due to the lack of assumptions regarding any underlying population. In our paper, we present a method that is distribution-free, making no assumptions about the MRF instances fed into our model.
> For further reading on non-parametric statistics, one can refer to the following resources:
> - Non-parametric statistics - Wikipedia[https://en.wikipedia.org/wiki/Nonparametric_statistics]
> - "Nonparametric Methods: The History, the Reality, and the Future (with Special Reference to Statistical Selection Problems)," page 63-83, available at: Springer Link.[https://link.springer.com/chapter/10.1007/978-3-642-46893-3_7]
>
> **W3**: We apologize for the misleading wording used in the abstract, where we stated that we outperform the baselines in all cases. We actually outperform them in nearly all cases, and we will clarify this in the final version of the paper. In our experiments, we conducted a qualitative analysis focused on the number of iterations to compare convergence speed. The reason we did not compare execution time is that we are utilizing unoptimized Python code, while the baselines are implemented with fine-tuned C++ code even over decades. Therefore, directly comparing the execution times would not be fair.
>
> **W4**:Sorry for that, here is the anonymous link to the code of our model.
> https://anonymous.4open.science/r/NeuroLifting-Neural-Inference-on-Markov-Random-Fields-at-Scale-160C
> You are welcome to run the code and we look forward to your further suggestions.
> Since I'm not quite familiar with Lifting and MRFs, I will keep a low confidence score.

---

> ### Comment · Reviewer_HyCv · 2024-11-24
>
> 1. Regarding W1, could you please theoretically justify your statement "we utilize the transformed problem to approximate the original one"? It remains unclear how the alternative GNN optimization problem effectively approximates the original problem. Can you present any analysis, such as an approximation error bound or formal guarantees, to support the validity of the advocated NeuroLifting framework?
>
> 2. I disagree with the authors' claim that the GNN used in this manuscript is a non-parametric model. GNNs do make the assumption that the data are drawn from an unknown parametric distribution; see e.g., [1, 2]. A parametric model is a family of probability distributions that has a finite number of parameters [3]. The architecture of a GNN is fixed and independent of the dataset size, and thus belong to parametric models.
>
> 3. [Toulbar2](https://github.com/toulbar2/toulbar2) also provides corresponding Python3 APIs. It is indeed unfair that no numerical tests are conducted to state that NeuroLifting "outperforms all existing approximate inference strategies in terms of solution quality without sacrificing computational efficiency".
> > The reason we did not compare execution time is that we are utilizing unoptimized Python code
> >
> Why don't you optimize your Python codes? This rationale does not justify omitting such a crucial comparison.
>
> 4. Thanks for the link, which addressed my concerns.
>
> [1] S. Fan, X. Wang, C. Shi, P. Cui and B. Wang, "Generalizing Graph Neural Networks on Out-of-Distribution Graphs," T-PAMI.
> [2] Y. Qin, X. Wang, Z. Zhang, P. Xie, W. Zhu, "Graph Neural Architecture Search Under Distribution Shifts", ICML.
> [3] Parametric model - [Wikipedia](https://en.wikipedia.org/wiki/Parametric_model).

---

> > ### Author Response · Authors · 2024-11-25
> >
> > **Q1**: We thank reviewer HyCv for the valuable feedback, which could help significantly improve the quality of our paper. The followings are our responses to your further concerns.
> >
> > **Approximation**: We want to emphasize that the original MRF inference problem on discrete variables (Eq. 1) is approximated with a soft probability relaxation (Eq. 5 and 6) like the common practice in [1, 2] which has been well studied in terms of both theoretical and empirical aspects. That said, until Eq. 5 and 6, we did not introduce any GNN concepts to approximate Eq. 1 yet. Instead, GNN is then employed to lift the relaxed problem of Eq. 5 and 6 into higher dimensions, aiming to ease the optimization using gradient backpropagation.
> >
> > To be clear, NeuroLifting consists of two parts of **"Probabilistic approximation + GNN optimization"**. The **approximation analysis** can directly inherit those from [3,4] (the validity also derive from these papers), while the **optimization aspects** largely align with GNNs [5,6] or even other neural network architechtures [7,8]. Establishing a unified theoretical framework incorporating both stands for a significant milestone, which is apparently beyond the scope of our paper. Instead, our purpose is to propose a **novel method** to perform MRF inference at scale.
> >
> > [1] Martin J. A. Schuetz, J. Kyle Brubaker, and Helmut G. Katzgraber. Combinatorial optimization with
> > physics-inspired graph neural networks. Nature Machine Intelligence, 4(4):367–377, April 2022.
> >
> > [2] Nair V, Bartunov S, Gimeno F, et al. Solving mixed integer programs using neural networks[J]. arXiv preprint arXiv:2012.13349, 2020.
> >
> > [3] Wang H P, Wu N, Yang H, et al. Unsupervised learning for combinatorial optimization with principled objective relaxation[J]. Advances in Neural Information Processing Systems, 2022, 35: 31444-31458
> >
> > [4] Karalias N, Loukas A. Erdos goes neural: an unsupervised learning framework for combinatorial optimization on graphs[J]. Advances in Neural Information Processing Systems, 2020, 33: 6659-6672.
> >
> > [5] Huang K, Zhai J, Zheng Z, et al. Understanding and bridging the gaps in current GNN performance optimizations[C]//Proceedings of the 26th ACM SIGPLAN Symposium on Principles and Practice of Parallel Programming. 2021: 119-132.
> >
> > [6] Xie Y, Li S, Yang C, et al. When do gnns work: Understanding and improving neighborhood aggregation[C]//IJCAI'20: Proceedings of the Twenty-Ninth International Joint Conference on Artificial Intelligence,{IJCAI} 2020. 2020, 2020(1).
> >
> > [7] Zhang C, Bengio S, Hardt M, et al. Understanding deep learning (still) requires rethinking generalization[J]. Communications of the ACM, 2021, 64(3): 107-115.
> >
> > [8] LeCun Y, Bengio Y, Hinton G. Deep learning[J]. nature, 2015, 521(7553): 436-444.
> >
> > **Q2**: We suppose reviewer HyCv has a fundamental misunderstanding about the statistical context of "non-parametric inference". **"Non-parametric" is to describe data, indicating the potential distribution of the data can be arbitrary.** The reasons why the model is non-parametric are as follows:
> >
> > 1.  When we talk about the training of our model, we are actually talking about the inference process. In other words, in NeuroLifting, "training" and "inference" are equivalent (both in a gradient-backpropagation fashion). When a new instance arrives, the model will be trained/start to infer after initialization. Which means we performed optimization on each problem **regardless of their distributions**. This, in a statistical sense, is non-parametric.
> >
> > 2. In the paper, we fixed the GNN settings for our tests to demonstrate that a moderately sized GNN can yield fairly good results. However, this does not imply that these parameters are inherently fixed. In Table 6 of Appendix B, we present the full results of our model on UAI pairwise instances with networks of various sizes. You will see that tuning model parameters can lead to enhanced performance.

---

> ### Author Response · Authors · 2024-11-25
>
> **Q3**: Even though Toulbar2 has Python 3 APIs, its backend code was written in C++ and has been kept optimized for decades. It means that the efficiency of Toulbar2 is attributed to the persistent contribution of researchers, programmers, and developers for a long period of time.
>
> In our time comparison, we conducted tests on the inference times using Toulbar2 with the PCI instances, and the results are presented in Table 7, which you can find in our latest response to reviewer qPjP. We demonstrated improvements in both time efficiency and solution quality **at scale**, as reflected in the trends shown in the results.
>
> The primary goal of our work is to validate the effectiveness of our method. While time efficiency is indeed a critical aspect, it is also a key focus for future development of our NeuroLifting method.
>
> However, your advice on this point is valuable to us. The method clearly demonstrates the potential for efficient inference in large-scale problems, and there is significant room for further improvements in its efficiency. We will revise our manuscript to provide a more thorough discussion and presentation regarding the efficiency of our approach.

---

### Official Review · Reviewer_fwQZ · 2024-11-02

**Soundness:** 3
**Presentation:** 3
**Contribution:** 3
**Rating:** 6
**Confidence:** 2

**Summary:**

The paper introduces NEUROLIFTING, a novel inference method that leverages Graph Neural Networks (GNNs) to optimize decision variables in large-scale MRFs. in contrast traditional methods, such as belief propagation and Toulbar2, which struggles to balance efficiency and solution quality as MRFs scale, NEUROLIFTING reparameterizes MRF variables with GNNs which enables efficient optimization using gradient descent while taking advantage of the smooth neural network landscape. This approach shows competitive results against exact solvers on smaller problems and superior performance on larger ones, with linear computational complexity growth.

**Strengths:**

* NEUROLIFTING introduces a new way to approach MRF inference by incorporating GNNs, enhancing scalability and parallelism compared to traditional methods.

*  Empirical results indicate that NEUROLIFTING consistently outperforms approximate inference methods in terms of solution quality and closely matches exact solvers on smaller instances.

*  The model shows linear growth in computational complexity, making it well-suited for large-scale MRFs.

* The method effectively transforms high-order graphs into a GNN-compatible format, expanding its applicability across various MRF-based problems.

**Weaknesses:**

* The use of GNNs, especially for large graphs, may still introduce significant computational costs and memory requirements, potentially limiting applicability in extremely large or resource-constrained environments.

* The effectiveness of NEUROLIFTING could be sensitive to the choice of GNN architecture, number of layers, and feature dimensions, which might require extensive tuning.

* Although NEUROLIFTING outperforms existing methods on large problems, there may be cases where it still falls short of the accuracy provided by exact solvers, especially on smaller and well-structured graphs.

**Questions:**

1. How sensitive is NEUROLIFTING's performance to different GNN architectures and hyperparameters? Can it be generalized across various MRF problems without extensive tuning?

2. How does the model perform on MRF instances with unique or less common graph structures that differ significantly from those used in the empirical tests?

3. Are there alternative ways to optimize the energy function or refine the padding strategy to further enhance efficiency or solution quality?

4. Could the NEUROLIFTING framework be extended to handle dynamic graphs or incorporate temporal information for applications in time-varying MRFs?

---

> ### Author Response · Authors · 2024-11-15
>
> W stands for weakness, Q stands for question
>
> **W1**: From the magnitude of instances in the UAI dataset and the real-world PCI instances, it is evident that the common MRF instances are not as large as those we have generated. For problems of this scale(up to 10k nodes with average degree 10), a single NVIDIA GeForce RTX 3090 24Gis capable of handling the computations effectively. In contrast, for larger-scale problems (50k nodes with average degree 10 or 100k nodes average degree 5), a single A100 80G is sufficient to manage the increased demands.
> Additionally, it's important to note that our current implementation of the code is written in Python (for research purpose) and has not been fully optimized, leaving significant room for further improvements. With targeted optimizations, we anticipate even better efficiency and scalability in handling these instances.
>
> We believe that a neural-network-driven optimization framework for traditional MRFs is promising for even larger scale problems, especially with the fast development of modern AI chips.
>
> **W3**:The primary motivation of our work is to address problems **at scale**, which are generally difficult for any traditional methods to our best knowledge. Such large-scale MRF problems broadly existing in the real world, but we are yet far from solving them nicely. This is why we include very dense graphs and graphs with a large number of nodes in the experiments conducted in Sections 4.1 and 4.3. For less challenging small problems beyond our main focus, existing solvers can readily be applied. The capability and flexibility of our model are well-suited to meet the large-scale needs of the current generation.
>
> **Q1/W2**: In all the experiments presented in Sections 4.1 and 4.3, we utilized 5-layer GNNs with a maximum of 150 iterations, while Section 4.2 employed 8-layer GNNs with a maximum of 100 iterations. The other settings were kept consistent across all experiments. Our ablation study in Section 4.4 illustrates the performance differences using various GNN backbones, showing that the primary variation arises from **convergence speed** rather than **final results**. We also experimented with different initialization strategies for the network parameters, but they did not significantly affect the outcomes, which is why those details are not included in the main paper. This suggests that our method is not particularly sensitive to these factors.
>
> Furthermore, our approach can be generalized to various MRF problems without extensive tuning, as demonstrated in Sections 4.1 and 4.3, where we applied the same settings to tackle MRFs of differing sizes, densities, and energy formulations.
>
> **Q2**:We are making significant efforts to encompass a wide variety of graph types in our experiments. The performance tests for our method utilize cases from the UAI dataset, which includes a diverse range of MRF problems and represents a commonly accepted dataset in the community. For the generated problems, we experimented with Erdős–Rényi (ER) graphs of varying densities and different energy functions. Additionally, we incorporated **real-world PCI problems**, which feature distinct graph structures, alongside the generated problems and those from the UAI dataset. We are continuously working to acquire more real-world problems to further evaluate the performance of our method.
>
> **Q3**: As discussed in Section 3.2 of our paper, we attempted to mask the padding portion and explored various padding strategies, including using infinities for padding. Unfortunately, none of these methods were able to produce a feasible solution. Our evaluations revealed that our current padding strategy has emerged as the most effective and suitable approach to be equipped with GNNs. The computation related to energy or loss is a challenging engineering problem, and addressing it would indeed be beneficial for enhancing the performance of our method.
>
> **Q4**: This is indeed an excellent question and presents a valuable avenue for future work. We can define the energy function as $E(X) = \sum \theta_t(C)$. In this context, we differentiate between two types of time-varying MRFs. The first type features only minor changes in the  $\theta_t$  parameters at each time step. In these scenarios, our method can be effectively applied, since MRFs are defined on local structures; thus, a small change in a specific part of the graph can significantly influence other areas. This enables us to leverage previous results as starting solutions.
>
> Conversely, when a majority of the $\theta_t$  parameters are changing—regardless of whether these changes are smooth or abrupt—we will need to retrain the model, as the objective function itself is also evolving. Currently, our main focus is on static MRFs.

---

> > ### Comment · Reviewer_fwQZ · 2024-11-20
> >
> > Thank you for your response. I don't have other questions. I keep my initial score due to my lack of certainty.

---

> > > ### Author Response · Authors · 2024-11-21
> > >
> > > Thank you very much for your insightful and encouraging feedback! We are delighted that our response has addressed your concerns regarding the paper. Your comments about our interesting findings are highly appreciated and have significantly helped us better present the core innovations of our work.

---

### Official Review · Reviewer_jC41 · 2024-11-09

**Soundness:** 3
**Presentation:** 2
**Contribution:** 2
**Rating:** 3
**Confidence:** 5

**Summary:**

The authors propose a GNN based approach to approximate MAP inference in MRFs.  The contribution definitely follows recent trends in trying to use NNs to approximately solve hard problems, it falls a bit short in its treatment of historical work and references to modern approaches.  Overall, the presentation needs quite a bit of improvement.

**Strengths:**

- The proposed approach is a novel combination of optimization ideas with modern NN approaches.

- The authors provide a wide array of experimental results across different domains and both synthetic and real data.

**Weaknesses:**

- Some of the text looks like it was fed into an MRF for "improvement," but it reads a bit awkwardly and uses imprecise statements.  Even if this isn't the case, I would suggest tidying it up a bit so as to make it read more like a traditional technical paper.

- The descriptions of MRFs and related work is poor.  Some parts of the discussion make it seems like some of the ideas being presented are novel to the community when they aren't really.  For example, the authors define MRFs factorizing over the cliques of a graphs and then claim novel insights that these cliques can be subdivided into edges when representing the graph.  This is, of course, what it means to be a clique in a graph.  Equation (4) isn't the standard loopy BP representation where inference is no longer exact (it isn't even clear why you need this formula anyway).  Saying that message passing looks like GNNs when it is in fact the other way around by design. Among many others.

- Some of the decisions made in the setup, e.g., the padding seem like that could lead to poor results in practice.  There is a comment about this in the paper, but the chosen method seems significantly suboptimal to me especially when the range of values that each potential function can take is small.

**Questions:**

- Figure 1 is really difficult to understand.

- I don't love calling this approach NeuroLifting because the are already "lifted" methods in relational MRFs.  I'm not sure what else I would call this, but given the long history in that domain optimization lifting strategies are not the first things that come to my mind when I hear the name.  Equation (6) is quite similar to traditional MAP relaxations (some of which are actually cited).  Saying

- Can you further explain the issues that arise with padding?  It seems clear that padding essentially with infinities is the correct theoretical thing to do -- what is the practical issue here?

- Lots of references should probably be added.  Here's one that is somewhat related to help get you started:

Arya, S., Rahman, T. &amp; Gogate, V.. (2024). Learning to Solve the Constrained Most Probable Explanation Task in Probabilistic Graphical Models. <i>Proceedings of The 27th International Conference on Artificial Intelligence and Statistics</i>, in <i>Proceedings of Machine Learning Research</i> 238:2791-2799 Available from https://proceedings.mlr.press/v238/arya24b.html.

---

> ### Author Response · Authors · 2024-11-15
>
> W stands for weakness, Q stands for Question
>
> **W1**: In this paper, we propose a method for performing **MAP inference on large-scale Markov Random Fields (MRFs)** based on Graph Neural Network (GNN) modeling and gradient-based Belief Propagation (BP). In this sense, our goal aligns with optimizating energies over MRFs in its standard context. Unlike traditional methods that rely solely on BP, which empirically demonstrate suboptimal performance with large-scale clique-based MRFs, our approach effectively addresses this issue especially on complex MRF instances at scale. Our method is applicable to any problem instance, and it demonstrates superior performance on large-scale MRFs, as shown in Tables 3, 4, and 5. If any aspects of our writing have caused confusion, we will revise them in the final version of this paper.
>
> **W2**: We stated in the manuscript that MRFs can be factorized over both cliques and edges because we aim to explain the implementation details of our method rather than claim our novelty on this part. Empirically, problems defined on cliques tend to be more challenging to solve than those defined on edges. In Sections 4.1 and 4.2, we test our method on both pairwise MRFs and high-order MRFs to demonstrate its applicability to both types.
>
> In essence, Equation (4) illustrates how LBP updates its node beliefs. We include this formula to highlight the message-passing scheme in LBP, which is an analogy in the message passing of GNNs. While there are significant differences in the message aggregation methods used in GNNs and LBP, within the LBP framework, nodes communicate by sending and receiving messages to and from their neighbors simultaneously in a parallel manner, akin to the approach employed in GNNs.
>
> **Q1**: In Section 3.1, we provide a brief explanation of the figure, and here we would like to offer a more detailed version. Figure 1 illustrates the graph in both clique graph form and edge graph form, using a six-node graph that consists of three cliques as an example. The energy function displayed above the figure indicates that we only have clique energy in this case.
> Next, the graph is input into the Neural Networks. The Multi-Layer Perceptron (MLP) encodes the nodes, after which the graph passes through the Graph Neural Network (GNN). The final output is processed by a softmax layer. After obtaining the output, we round the network outputs to derive a configuration. Subsequently, we update the network parameters to minimize the loss function, such as the energy function.
>
> We will revise the figure to ensure that it clearly conveys our ideas in the final version.
>
> **Q2**: While lifting has indeed been established in the MRF domain, as briefly mentioned at the end of Section 2, this is not our main claim in this paper. However, we are the first to propose the application of Neural Networks for lifting to MRFs using multiple GNN layers, whereas existing works have primarily focused on lifting from a traditional optimization perspective (not a stack of GNN layers). This is why we refer to our approach as Neurolifting
>
> The concept of the loss function is based on the traditional relaxation method of optimization. However, this approach may not be effective if we do not adequately address the challenges associated with high-dimensional problems and inconsistent dimensionalities. In the cited paper, the authors focus on pairwise relationships, which are relatively straightforward to implement.
>
> In short, NeuroLifting seeks to lift original MRF problems with extra optimizable variables in GNN layers.
>
> **Q3/W3**: In the field of machine learning, padding is a widely adopted technique that serves practical purposes in model training and data processing. Rather than being solely rooted in theoretical considerations, padding addresses various engineering challenges that arise during model implementation.
>
> In our experiments, we explored multiple padding strategies, including the approach of padding with infinities. While we discovered that our current padding strategy emerged as the most effective solution based on our evaluations. We have provided an explanation of our chosen method in the Remarks section of 3.2.
>
> Additionally, we would like to propose another possible issue associated with padding using infinities. Specifically, when infinities are employed as padding values, even minor fluctuations in the neural network's output can lead to significant variations in the loss function. This sensitivity can complicate the optimization process, making convergence more difficult. Of course, this instability originates from neural network itself.
>
> We anticipate that this topic warrants a more thorough exploration, and we plan to include a detailed discussion on this matter in the final version of our paper.

---

> ### Author Response · Authors · 2024-11-15
>
> **Q4** :In the paper you mentioned, the authors propose a self-supervised model capable of solving MPE (Most Probable Explanation) problems, similar to our approach. However, our perspectives are fundamentally different. Their motivation stems from the Learning to Optimize (L2O) method, which primarily involves inputting a problem instance into a neural network (NN) that then outputs a solution. In contrast, we employ Graph Neural Networks (GNNs) to perform the lifting process.
>
> This distinction also leads to significant differences in our loss functions. Their loss function derives from traditional optimization techniques, specifically utilizing Lagrange multipliers to formulate an unconstrained problem. Our approach, however, is more straightforward; we ensure that the constraints inherent to Markov Random Fields (MRFs) are met through a carefully designed padding and rounding scheme. A detailed explanation of this aspect can be found in response to question 10 from reviewer qPjP.
> Moreover, our model is applicable to a broader range of MRFs, including high-order MRFs, while the authors of the other paper only test on smaller instances, all of which are restricted to pairwise MRFs.
>
> We can include a discussion of these differences in the final version of our paper and have more papers in the related work.

---

> ### Author Response · Authors · 2024-11-25
>
> Dear Reviewer  jC41,
>
> We appreciate your valuable time and thorough review. Our rebuttal is concise due to space limitations. However, if you have any remaining concerns or questions, we are more than willing to provide additional clarification. We greatly value your input and would appreciate any further suggestions or comments you may have regarding enhancing our manuscript.
>
> Best regards.

---

### Author Response · Authors · 2024-11-15

Table 4 High-order cases of UAI
|Graph | Neurolifting Loss | Energy |
|---------------------------------------------------------------------------------|--------------------------------------|------------------------------------|
|Maxsat\_gss-25-s100   |-139904.266 $\pm$ 1717.483 | -139914.341 $\pm$ 1710.139 |
|BN-nd-250-5-10        | 189.395 $\pm$ 5.424 | 187.729 $\pm$ 4.966 |
|Maxsat\_mod4block\_2vars\_10gates\_u2\_autoenc         | -146166.620 $\pm$ 41250.035 | -146166.797 $\pm$ 41249.859|
|Maxsat\_mod2c-rand3bip-sat-240-3.shuffled-as.sat05-2520      | -3511.994 $\pm$ 220.654 | -3511.822 $\pm$ 220.471 |
|Maxsat\_mod2c-rand3bip-sat-250-3.shuffled-as.sat05-2535       |-3686.567 $\pm$ 166.998 | -3686.085 $\pm$ 166.499 |

Table 5 PCI cases with error
|Graph | Neurolifting Loss | Energy |
|-------------------|-----------------------------|-------------------------------|
|PCI\_1             |  18.851 $\pm$ 0.101 |   18.841 $\pm$ 0.095 |
|PCI\_2             |  100.167 $\pm$ 0.685 | 100.151 $\pm$ 0.693 |
|PCI\_3             |  1010.757 $\pm$ 1.665 | 1010.687 $\pm$ 1.637 |
|PCI\_4             |  421.030$\pm$ 5.650| 420.734 $\pm$ 5.356 |
|PCI\_5             |  1109.976 $\pm$ 22.260 | 1108.144 $\pm$ 20.298 |

Table 6 GNN loss using different parameter initialization
|Graph Type |    Uniform | Kaiming | Xavier |
|--------------------|-------------------------------------------------|---------------------------------------------------|------------------------------------------------|
|Synthetic data |    -62706.173 $\pm$ 12540.261          |     -89331.936 $\pm$ 1059.104           |   -89276.337 $\pm$ 1117.219           |
|UAI                    |     11618.903 $\pm$ 25179.166         |        -39010.959 $\pm$ 372.373        |       11618.903 $\pm$ 25179.166       |
|PCI                    |      3045.851 $\pm$ 202.735        |       2642.396 $\pm$ 9.121        |     2643.387 $\pm$ 8.315       |

---

### Author Response · Authors · 2024-11-15
**General Response**

The tables below correspond to their respective tables in the paper, presenting the loss from the loss function we designed alongside the corresponding energy values. Each box includes the standard error to illustrate the sensitivity of our model to the different initializations. As evident from these results, the discrepancy between the loss and the actual energy is minimal. Moreover, almost all the energy errors are below 1% when compared to the average energy values.

Table 6 presents the performance of our model using three different initialization methods: uniform initialization, Kaiming initialization, and Xavier initialization. As observed, the performance with both Kaiming and Xavier initializations is quite similar. Since both methods are commonly used, we did not emphasize this finding in our paper.

Here is the anonymous link to the code of our model.
https://anonymous.4open.science/r/NeuroLifting-Neural-Inference-on-Markov-Random-Fields-at-Scale-160C

Table 1 synthetic pariwise cases with error

|Graph          | Neurolifting loss| Energy |
|-----------------| --------------------------------------|--------------------------------------|
|P_potts_1|      -21791.868 $\pm$ 218.106  | -21799.268 $\pm$ 216.075 |
|P_potts_2|      -105762.092 $\pm$ 434.674    | -106016.855 $\pm$ 168.861 |
|P_potts_3|      -211406.914  $\pm$ 1489.099    |  -210182.681 $\pm$ 275.164 |
|P_potts_4|      12219.817  $\pm$ 538.969  | 11811.682  $\pm$ 123.877  |
|P_potts_5|      12010.036 $\pm$ 266.610    | 11673.708 $\pm$ 146.513  |
|P_potts_6|      18399.913 $\pm$ 1475.526    | 16988.347 $\pm$ 163.587 |
|P_potts_7|      17480.701 $\pm$ 212.084    | 17265.434 $\pm$ 140.904  |
|P_potts_8|      26115.840 $\pm$ 1677.627   |  24668.087 $\pm$ 163.587 |
|P_potts_9|      26348.525 $\pm$ 1095.319   | 25189.789 $\pm$ 132.693 |
|-----------------| --------------------------------------|------------------------------------- |
|P_random_1 | -4570.079 $\pm$ 31.228 | -4574.664 $\pm$ 31.411|
|P_random_2 | -21774.416 $\pm$ 52.910 | -21798.702 $\pm$ 32.389 |
|P_random_3 | -41953.991  $\pm$ 237.577 | -41972.379 $\pm$ 216.574 |
|P_random_4 | 12552.252 $\pm$ 30.311 | 11983.388$\pm$ 213.454 |
|P_random_5 | 12308.580 $\pm$ 14.045 | 11945.450 $\pm$ 194.481 |
|P_random_6 | 17705.219 $\pm$ 435.560  | 17207.997 $\pm$ 405.217 |
|P_random_7 | 18343.026 $\pm$ 1448.821 | 16971.435 $\pm$ 209.021 |
|P_random_8 | 25949.446 $\pm$ 995.956 | 24787.343 $\pm$ 163.587 |
|P_random_9 | 25871.264 $\pm$ 1087.915 |  24811.354 $\pm$ 171.315 |

Table 2 synthetic high-order cases with error
|Graph | Neurolifting Loss | Energy |
|-----------------| --------------------------------------|------------------------------------- |
|H_Instances_1 |   -29844.092 $\pm$ 450.184 | -29854.202 $\pm$ 450.079 |
|H_Instances_2 |   -20260.289 $\pm$ 143.276 | -20286.571 $\pm$ 143.624 |
|H_Instances_3 |   -3432.3205 $\pm$ 9.520| -3611.379 $\pm$ 9.654 |
|H_Instances_4 |   9975.517 $\pm$ 80.854 | 9754.885 $\pm$ 27.806 |
|H_Instances_5 |   10403.888 $\pm$ 798.812 | 10217.842 $\pm$ 848.029 |

Table 3 UAI cases with error
|Graph | Neurolifting Loss | Energy |
|-----------------| --------------------------------------|------------------------------------- |
|ProteinFolding_11 | -3976.908 $\pm$ 52.047 | -4018.784 $\pm$ 36.491 |
|ProteinFolding_12 | 16137.682 $\pm$ 16.020 | 16090.801 $\pm$ 22.869 |
|Grids_19                 | -2400.251 $\pm$ 20.061 | -2398.078 $\pm$ 16.010 |
|Grids_21                 | -16592.926 $\pm$ 94.368 | -16605.564 $\pm$ 113.096 |
|Grids_24                 | -16323.767 $\pm$ 171.950 | -16222.104 $\pm$ 222.593 |
|Grids_25                 | -2361.900 $\pm$ 10.231 | -2361.055 $\pm$ 12.678 |
|Grids_26                 | -2595.041 $\pm$ 43.306 | -2577.378 $\pm$ 39.370 |
|Grids_27                 | -10898.595 $\pm$ 160.435 | -10771.257 $\pm$ 170.329 |
|Grids_30                 | -2651.035 $\pm$ 35.508 | -2676.246 $\pm$ 19.886 |
|Segmentation_11    | 432.291 $\pm$ 34.208 | 391.971 $\pm$ 40.099 |
|Segmentation_12    | 147.531 $\pm$ 26.215 | 128.759 $\pm$ 36.213 |
|Segmentation_13    | 147.845 $\pm$ 6.298 | 118.685 $\pm$ 34.864 |
|Segmentation_14    | 105.203 $\pm$ 7.717 | 103.269 $\pm$ 8.121 |
|Segmentation_15    | 417.276 $\pm$ 22.357 | 408.214 $\pm$ 27.037 |
|Segmentation_16    | 113.389 $\pm$ 4.81 | 108.415 $\pm$ 8.77 |
|Segmentation_17    | 494.82 $\pm$ 20.158 | 478.881 $\pm$ 42.455 |
|Segmentation_18    | 104.201 $\pm$ 2.692 | 97.555 $\pm$ 6.364 |
|Segmentation_19    | 96.173 $\pm$ 4.731 | 84.882 $\pm$ 10.063 |
|Segmentation_20    | 358.238 $\pm$ 34.148 | 320.027 $\pm$ 21.219 |

---

### Meta-Review · Area_Chair_xcCC · 2024-12-18

**Metareview:**

The paper introduces NEUROLIFTING, a technique using GNNs to optimize inference in large-scale MRFs. By reparameterizing decision variables and enabling gradient-based optimization, NEUROLIFTING achieves efficient, parallelizable performance. Empirical results show it matches exact solvers on moderate scales and outperforms the existing large-scale MRFs.

Although the paper's approach to inferring large-scale MRFs using GNNs is interesting, the reviewers highlighted the following main weaknesses:

1. One reviewer criticized the poor description of MRFs and related work, noting that some ideas presented as novel are already well-known in the community.
2. Another reviewer questioned the validity of the lifting method, highlighting the lack of theoretical justification for the equivalence between the original problem and the GNN-based optimization.
3. Several reviewers pointed out the absence of thorough experimental evaluation.
4. A reviewer challenged the claim that the proposed method is non-parametric.

**Additional Comments On Reviewer Discussion:**

Three reviewers recommended rejection, while one recommended a weak accept. The AC concurs with the reviewers' concerns and supports a recommendation for rejection. The AC also strongly encourages resubmission, recognizing the paper's importance and interest but acknowledging that it requires substantial revision—beyond what can be expected between initial submission and camera-ready submission.

---

### Decision · Program_Chairs · 2025-01-22

Reject